# A Simple Framework for Open-Vocabulary Zero-Shot Segmentation

**Thomas Stegmüller**[1]* **Tim Lebailly**[2]* **Nikola Đukić**[2]
**Behzad Bozorgtabar**[1,3] **Tinne Tuytelaars**[2] **Jean-Philippe Thiran**[1,3]

[1]EPFL [2]KU Leuven [3]CHUV

[1]{firstname}.{lastname}@epfl.ch [2]{firstname}.{lastname}@esat.kuleuven.be

## Abstract

Zero-shot classification capabilities naturally arise in models trained within a vision-language contrastive framework. Despite their classification prowess, these models struggle in dense tasks like zero-shot open-vocabulary segmentation. This deficiency is often attributed to the absence of localization cues in captions and the intertwined nature of the learning process, which encompasses both image/text representation learning and cross-modality alignment. To tackle these issues, we propose SimZSS, a **Sim**ple framework for open-vocabulary **Z**ero-**S**hot **S**egmentation. The method is founded on two key principles: *i)* leveraging frozen vision-only models that exhibit spatial awareness while exclusively aligning the text encoder and *ii)* exploiting the discrete nature of text and linguistic knowledge to pinpoint local concepts within captions. By capitalizing on the quality of the visual representations, our method requires only image-caption pair datasets and adapts to both small curated and large-scale noisy datasets. When trained on COCO Captions across 8 GPUs, SimZSS achieves state-of-the-art results on 7 out of 8 benchmark datasets in less than 15 minutes. Our code and pretrained models are publicly available at https://github.com/tileb1/simzss.

## 1 Introduction

Semantic segmentation stands as a cornerstone task within the realm of computer vision, playing a pivotal role in various applications ranging from autonomous driving to medical image analysis.

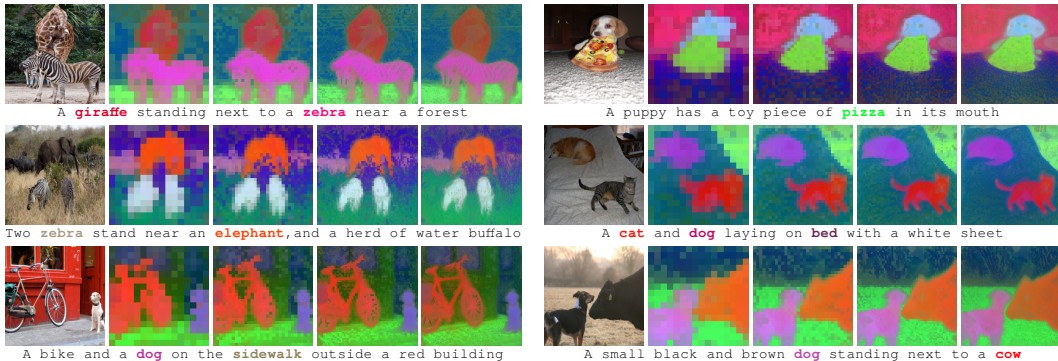

Figure 1: **Visualization of the patch-level representations and text concepts in the RGB color space.** PCA is used to map the dense representation of a single image into a three-dimensional space. The three-dimensional representations (color) of the text concepts from the concept bank and the corresponding caption are obtained and shown for each image. Each row includes the original image and dense feature visualization at different resolutions. These include the training resolution ($16 \times 16$) and higher resolutions ($2\times$, $4\times$, and $8\times$).

---

*denotes equal contribution.

However, its widespread adoption and scalability are hindered by the inherent label-intensive nature of traditional methods, demanding copious amounts of fine-grained annotations for training. Furthermore, traditional approaches, *e.g.*, Strudel et al. (2021); Cheng et al. (2022) are typically closed-vocabulary, meaning they only work for a pre-defined set of categories and generalize poorly to unseen classes. Self-supervised learning paradigms (Caron et al., 2021; Zhou et al., 2022b; He et al., 2020; Balazevic et al., 2024; Lebailly et al., 2024), which use pretext tasks to learn discriminative representations from data itself, offer a promising solution to alleviate the annotation burden. Representations obtained in this manner are typically clustered semantically, potentially even at a fine-grained level (Balazevic et al., 2024; Lebailly et al., 2024; Stegmüller et al., 2023) and, as such, yield excellent performance in various applications. Nonetheless, a degree of labeling is still necessary, whether for finetuning or constructing the support set of a $k$-NN classifier, for instance.

Recently, vision-language contrastive learning has proven to be a simple and effective approach for transforming web-scale amounts of noisy image-caption pairs into zero-shot classification capabilities (Radford et al., 2021; Jia et al., 2021). Nonetheless, the substantial computational and data requirements of these methods pose significant challenges. Furthermore, the performance of these methods on finer-grained downstream tasks, such as zero-shot segmentation tends to be subpar (see Tables 1 and 2). We argue that this is not only the consequence of a lack of local vision-language alignment but also a hint that the learned visual representations therein might be suboptimal. Based on this same observation, LiT (Zhai et al., 2022) proposed aligning a text encoder with a pretrained and frozen pure vision model. By decoupling the image representation learning process from the vision-language alignment process, this methodology enhances both computational and data efficiency while also improving performance in zero-shot classification tasks. The strong performance of LiT suggests that achieving zero-shot classification or segmentation relies on two key requirements:

1. The vision encoder must be capable of clustering images or groups of pixels with shared semantics.

2. The text encoder must be sufficiently aligned with the visual modality to accurately label these clusters.

The moderate performance of frozen vision encoder pretrained with CLIP on pure vision tasks (see Appendix A.3) combined with the stellar results of LiT support that the second requirement is significantly easier to accomplish than the first one and that the first requirement can be addressed independently of the second. Fortunately, various studies (Hamilton et al., 2022; Siméoni et al., 2021; Wang et al., 2023; Zhang et al., 2023; Siméoni et al., 2023) have observed and capitalized on the spatial awareness of self-supervised pretrained vision transformers (ViTs). These works have demonstrated remarkable capabilities for dense downstream tasks with minimal or even zero trainable parameters (Balazevic et al., 2024). This evidence suggests that self-supervised vision transformers may just be one text encoder away from becoming open-vocabulary segmenters. We investigate this hypothesis and propose a simple framework for open-vocabulary zero-shot segmentation. In essence, we capitalize on a pretrained and frozen vision encoder that exhibits excellent fine-grained semantic coherence and focus solely on aligning a text encoder to it. To effectively enforce a localized consistency objective, we leverage linguistic knowledge to identify concepts in captions and build on the quality of the frozen vision encoder to retrieve the corresponding concepts in images.

Our main contributions are as follows:

(i) SimZSS is designed to be simple, with minimal hyperparameters, making it highly compatible with both small curated datasets and large-scale noisy datasets.

(ii) The proposed framework is robust, supporting various pretrained vision towers, and accommodating both supervised and self-supervised pretraining of the vision backbone.

(iii) By decoupling visual representation learning from cross-modality concept-level alignment, our proposed framework, SimZSS achieves high computational and data efficiency.

## 2 RELATED WORKS

**Open-vocabulary learning.** Traditionally, computer vision methods operate under the closed-vocabulary hypothesis. This assumption presumes that all object categories a model is expected to classify, detect, or segment at the test time are already known and labeled during training. This presents significant challenges due to the extensive labeling required and the limited generalizability of the resulting models. Open-vocabulary learning aims to eliminate these limitations. Notably, this is a more challenging objective than the one targeted by self-supervised visual representation learning (Chen & He, 2021; Caron et al., 2021; Li et al., 2021; Grill et al., 2020; Oquab et al., 2023), which assumes the availability of labels at test time. To address this additional constraint, a text encoder is typically trained jointly with the vision tower to maximize vision-language alignment on large amounts of image-caption pairs via contrastive learning (Radford et al., 2021; Jia et al., 2021). Subsequent studies have either focused on improving the cross-modality alignment pretext task (Alayrac et al., 2022; Yu et al., 2022; Tschannen et al., 2024) or on improving the computational efficiency (Zhai et al., 2022; Li et al., 2023b). However, while these models excel at classification, their performance on dense downstream tasks is usually subpar.

**Open-vocabulary segmentation.** The shortcomings of open-vocabulary methods on dense downstream tasks have been the focus of significant research efforts. A recurring approach in the literature is to use dense annotations for learning segmentation and to leverage a pretrained text encoder to provide the weights of a generalizable classifier (Li et al., 2022; Ghiasi et al., 2022; Liang et al., 2023; Ding et al., 2022; Zhou et al., 2022a; Xu et al., 2023b; Ma et al., 2022; Yu et al., 2024). While these methods demonstrate excellent zero-shot segmentation performance, they only partially alleviate the pixel-level annotation requirements.

To overcome this limitation, approaches that do not require segmentation masks have been proposed. PACL (Mukhoti et al., 2023) demonstrates the effectiveness of training only the projections at the interface of the two modalities and bootstrapping patch-level cross-modal similarities. Similarly, TCL (Cha et al., 2023) leverages existing fine-grained similarities to jointly learn text-grounded masks and impose contrastive region-text alignment based on the obtained masks. Alternatively, GroupViT Xu et al. (2022) proposes a specialized architecture that performs hierarchical grouping/pooling of visual tokens based on their similarity with learnable query tokens. Another line of work showed the benefits of integrating a consistency objective between augmented and/or masked views of the input image (Dong et al., 2023; Ren et al., 2023) in the context of vision-language contrastive learning. Recent works (Shin et al., 2022; Sun et al., 2024) also showed the possibility of achieving excellent zero-shot segmentation capabilities without any additional training. Finally, Wysoczańska et al. (2024; 2023); Rewatbowornwong et al. (2023) combine the vision-language alignment capabilities of CLIP (Radford et al., 2021) with the spatial-awareness of self-supervised vision transformers, *e.g.*, DINO (Caron et al., 2021) to develop open-vocabulary zero-shot segmenter.

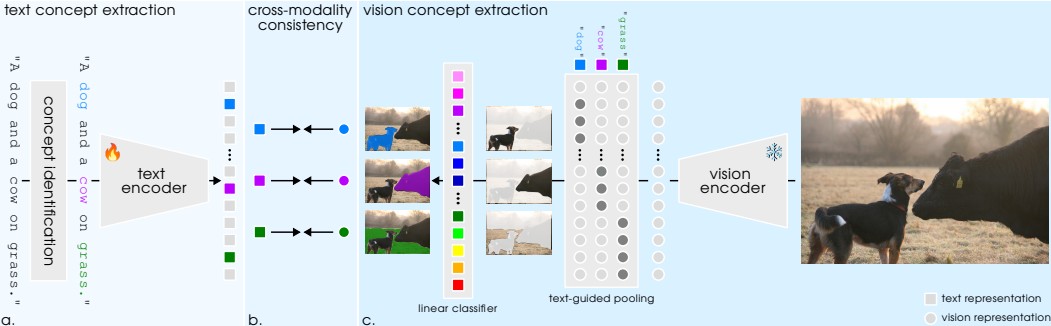

Figure 2: **Overview of SimZSS**. On the text side (**a.**), each concept in the caption is represented using a trainable text encoder. On the vision side (**c.**), visual representations of each concept are obtained via a similarity-based pooling of the visual tokens. These visual concept representations are then projected onto a linear classifier, with weights derived from the text concept representations of the current batch. Cross-modality consistency is enforced using cross-entropy loss (**b.**).

## 3 METHOD

### 3.1 PRELIMINARIES

We first provide a brief primer on the terminology used throughout the paper.

**Dense representation.** Transformers encode an input signal as a sequence of tokens, and we refer to the corresponding output sequence as the *dense representation*. For vision transformers, this corresponds to a tensor $\boldsymbol{z}_v \in \mathbb{R}^{n_v \times d_v}$, where $d_v$ is the dimension of the representation space and $n_v$ is the number of patches in the input image. For text input, the dense representation $\boldsymbol{z}_t \in \mathbb{R}^{n_t \times d_t}$ is a sequence of $d_t$-dimensional tokens.

**Local representation.** We refer to the result of indexing a dense representation with a sequence index $i$ as a *local representation*. This translates to $\boldsymbol{z}_t^i \in \mathbb{R}^{d_t}$ for textual input signals and $\boldsymbol{z}_v^i \in \mathbb{R}^{d_v}$ for visual input signals.

**Concept representation.** Aggregating local representations corresponding to the $l^{th}$ semantic concept in the input signal yields a *concept representation*. For visual input, this is denoted as $\mathbf{c}_v^l \in \mathbb{R}^{d_v}$, whereas for textual input, we use $\mathbf{c}_t^l \in \mathbb{R}^{d_t}$.

**Global representation.** A representation of the entire input signal is referred to as the *global representation*. Specialized tokens are typically used for this purpose. In vision transformers, the global representation $\bar{\boldsymbol{z}}_v \in \mathbb{R}^{d_v}$ is the [CLS] token, while for a text encoder $\bar{\boldsymbol{z}}_t \in \mathbb{R}^{d_t}$ denotes the [EOS] token.

### 3.2 VISION & LANGUAGE CONCEPT IDENTIFICATION

At a conceptual level, enforcing cross-modality consistency faces a significant challenge due to the complexity of identifying and matching concepts across modalities. In a nutshell, our proposed solution leverages the discrete nature of textual data, enabling the straightforward identification of concepts within captions. Subsequently, we retrieve associated visual concepts, conditioned on those identified within the text, effectively circumventing the complexity of cross-modality concept matching.

#### 3.2.1 TEXTUAL CONCEPT IDENTIFICATION

Given an image-caption pair $(\boldsymbol{x}_v, \boldsymbol{x}_t)$, we utilize the text modality to identify semantic concepts. Language adheres to structured grammatical rules, allowing us to leverage linguistic knowledge to pinpoint the main subjects and objects within a sentence. Since captions are intended to describe images, the primary subjects and objects in the captions are likely to correspond to visual elements in the images. Therefore, we employ a part-of-speech (POS) tagger, which automatically classifies words according to their grammatical roles. In our context, noun phrases (NPs) are of particular interest, as they generally encapsulate the main subjects and objects in the sentences, providing a direct link to the visual concepts depicted in the images. However, captions can be quite noisy, so further filtering is necessary to discard NPs that are unlikely to appear in the images. To address this, we refine a noun phrase into a concept by restricting it to the noun and its first compound. We then discard any concept that is absent in a predefined bank of concepts.

Once the concepts in the caption have been identified, it remains to obtain their equivalent in the representation space. This is accomplished by tracking the token indices spanned by each concept. Denoted as $\mathcal{S}_l$, this set represents the indices associated with the $l^{th}$ concept. The concept representation $\mathbf{c}_t^l \in \mathbb{R}^{d_t}$ is then derived by averaging the local representation at the corresponding indices:

$$\mathbf{c}_t^l = \frac{1}{|\mathcal{S}_l|} \sum_{i \in \mathcal{S}_l} \boldsymbol{z}_t^i \tag{1}$$

The $l^{th}$ textual concepts $\mathbf{c}_t^l \in \mathbb{R}^{d_t}$ is then mapped to the visual space via a linear projection $g$ : $\mathbb{R}^{d_t} \to \mathbb{R}^{d_v}$:

$$\tilde{\mathbf{c}}_t^l = g\left(\mathbf{c}_t^l\right) \tag{2}$$

In the remainder of this paper, we also refer to $\tilde{\mathbf{c}}_t^t$ as a text concept.

### 3.2.2 Visual concept identification

Upon identifying the $l^{th}$ text concept $\tilde{\mathbf{c}}_t^l \in \mathbb{R}^{d_v}$, we use it to query the dense visual representation and obtain the corresponding visual concept $\mathbf{c}_v^l \in \mathbb{R}^{d_v}$. To do this, we first embed the image $\boldsymbol{x}_v$ using a vision encoder $f_v$, which outputs the dense visual representation $\boldsymbol{z}_v \in \mathbb{R}^{n_v \times d_v}$ and a global visual representation $\bar{\boldsymbol{z}}_v \in \mathbb{R}^{d_v}$. We then compute the similarities between each local visual representation and the query text concept:

$$\mathbf{s} = \texttt{softmax}\left(\frac{\boldsymbol{z}_v \tilde{\mathbf{c}}_t^l}{\tau}\right) \tag{3}$$

where $\tau$ is a temperature parameter that regulates the sharpness of the similarity distribution (see Tab. 9). Consequently, the $l^{th}$ visual concept is obtained by bootstrapping existing cross-modality similarities, *i.e.*, via similarity-based pooling (see Fig. 2):

$$\mathbf{c}_v = \boldsymbol{z}_v^\top \mathbf{s} \tag{4}$$

This process is performed in parallel for all concepts from the $b$ captions in the batch, resulting in $\tilde{b}$ cross-modality concept pairs for which consistency can be encouraged.

### 3.3 Cross-modality consistency

In Section 3.2, we outline a methodology for identifying pairs of local concepts across textual and visual modalities. This method leverages existing similarities and thus may benefit from additional supervision at the beginning of training, as discussed in Section 3.3.1.

### 3.3.1 Global consistency

We now discuss the global consistency objective, which ensures similarities between images and captions. To this end, let $\bar{\boldsymbol{Z}}_v \in \mathbb{R}^{b \times d_v}$ be the matrix containing all global visual representations $\bar{\boldsymbol{z}}_v$ within a batch, and let $\bar{\boldsymbol{Z}}_t \in \mathbb{R}^{b \times d_t}$ be its equivalent in the text modality. After projecting the text concepts to the visual space, we can compute the global cross-modality similarity:

$$\bar{\mathbf{s}} = g\left(\bar{\boldsymbol{Z}}_t\right) \bar{\boldsymbol{Z}}_v^\top \tag{5}$$

The learning objective is to maximize the similarity of paired entries and minimize that of unpaired ones:

$$\mathcal{L}_g = -\frac{1}{2b} \sum_i \log\left(\frac{\exp\left(\bar{\mathbf{s}}_{ii}\right)}{\sum_j \exp\left(\bar{\mathbf{s}}_{ij}\right)}\right) - \frac{1}{2b} \sum_j \log\left(\frac{\exp\left(\bar{\mathbf{s}}_{jj}\right)}{\sum_i \exp\left(\bar{\mathbf{s}}_{ij}\right)}\right) \tag{6}$$

Overall the global consistency objective is identical to the one used in CLIP (Radford et al., 2021).

### 3.3.2 Concept-level consistency

After obtaining pairs of vision-language concepts, an intuitive approach to enforce consistency at the concept level is to use a contrastive objective, akin to the one described in Section 3.3.1, but applied between concepts from each modality. Empirically, we found that this approach did not yield the desired performance improvements on dense downstream tasks. At a global level, images and captions represent complex scenes, supporting the hypothesis that only $b$ positive image-caption pairs exist among the $b^2$ possible ones in a batch. Conversely, concepts typically encode individual objects that are likely to occur multiple times within a batch. This suggests that an instance-level objective is ill-suited for our setting. Therefore, we opt for a semantic-level objective.

Let's define $\mathbf{C}_t \in \mathbb{R}^{\tilde{b} \times d_t}$ representing the set of all $\tilde{b}$ text concepts in a batch and $\mathbf{C}_v \in \mathbb{R}^{\tilde{b} \times d_v}$ as its counterpart in the visual space. Thanks to the discrete nature of text, it is straightforward to keep track of concepts occurring in a batch, with each unique concept assigned a specific index, conveniently stored in $\mathbf{q} \in \{0, 1, ..., k-1\}^{\tilde{b}}$ (where $k$ denotes the number of unique concepts in a batch).

The weights $\mathbf{h} \in \mathbb{R}^{k \times d_v}$ of a linear classifier can be computed by summing the representations of identical concepts:

$$\mathbf{h}_i = \sum_j \mathbb{1}_{\{\mathbf{q}_j = i\}} g\left(\mathbf{C}_t\right)_j \tag{7}$$

After $l2$-normalizing the columns of both $\mathbf{h}$ and $\mathbf{C}_v$, we project one onto the other to derive a probability distribution:

$$\mathbf{p} = \underset{k}{\mathrm{softmax}}\left(\mathbf{C}_v \mathbf{h}^\top\right) \tag{8}$$

It follows that the cross-entropy loss can be used to ensure consistency between the query text concept and the retrieved visual concept:

$$\mathcal{L}_l = \frac{1}{\tilde{b}} \sum_i \sum_j -\mathbb{1}_{\{\mathbf{q}_i = j\}} \log\left(\mathbf{p}_{ij}\right) \tag{9}$$

The overall objective of SimZSS, denoted as $\mathcal{L}_{\mathrm{tot}}$, is a weighted sum of the global and local consistency objectives:

$$\mathcal{L}_{\mathrm{tot}} = \mathcal{L}_g + \lambda \mathcal{L}_l \tag{10}$$

where $\lambda$ is a weighting parameter whose impact is ablated in Table 9.

## 4 EXPERIMENTS

In this section, we investigate the properties of SimZSS through various experiments. Additional experiments can be found in Appendix A.

### 4.1 EXPERIMENTAL SETUP

#### 4.1.1 PRETRAINING DATASETS

We train our models on two distinct datasets: COCO Captions (Lin et al., 2014; Chen et al., 2015) and LAION-400M (Schuhmann et al., 2021). The former comprises 600K image-caption pairs with high-quality human-generated captions, while the latter is derived from LAION-5B (Schuhmann et al., 2022) consisting of image-caption pairs with vision-language cosine similarity exceeding 0.3, as determined by a pretrained CLIP model (Radford et al., 2021). These datasets represent opposing ends of the spectrum in terms of curation and scale.

#### 4.1.2 NETWORKS ARCHITECTURES

Vision transformers (Dosovitskiy et al., 2020) are used to obtain image representations. More precisely, we experiment with ViT-S/14 and ViT-B/14 pretrained with DINOv2 (Oquab et al., 2023) on LVD-142M. A ViT-B/16 pretrained with AugReg (Steiner et al., 2022) on ImageNet-21k is also tested. The architecture of the text transformers is identical to the one used in CLIP, and its weights are randomly initialized. Overall, the only architectural difference w.r.t. CLIP is the removal of the learnable linear layer that maps visual representations to the cross-modal representation space *i.e.*, we project textual representations directly onto the visual space rather than projecting both the textual and visual representations onto an intermediary space.

#### 4.1.3 OPTIMIZATION

For COCO Captions, we conduct training over 4M processed samples ($\sim 6.6$ epochs) using a global batchsize of 16,384. We incorporate a warm-up strategy spanning 10% of the training steps, linearly ramping up the learning rate until it reaches its peak value, chosen from the set {8e-5, 3e-5, 8e-6, 3e-6}[1]. Subsequently, we employ a cosine decay schedule for the remaining steps. Similarly, for LAION-400M, we train for 1 epoch with a global batchsize of 32,768, and we set the learning rate from the options {3e-5, 8e-6, 3e-6, 8e-7, 3e-7}. The remaining optimization settings align with those of OpenCLIP (Ilharco et al., 2021).

---

[1]scaled by `batchsize / 256`

### 4.1.4 TEXT CONCEPTS

We use the `en_core_web_trf` model from SpaCy (Honnibal et al., 2020) as part-of-speech tagger to identify noun phrases. The concept bank is obtained as the union of the class names from Pascal VOC (Everingham et al., 2012), Pascal Context (Mottaghi et al., 2014), COCO-Stuff (Caesar et al., 2018), Cityscapes (Cordts et al., 2016) and ADE20K (Zhou et al., 2017). This results in 574 concepts.

## 4.2 ZERO-SHOT SEGMENTATION OF FOREGROUND

Table 1: **Zero-shot foreground segmentation.** Pixel-wise predictions are obtained by projecting patch representations onto pre-computed text embeddings of the class names, followed by upsampling. The mIoU scores are reported across five standard segmentation datasets. † refers to our reproduction using DINOv2 pretrained vision backbones. The remaining results are as reported in Wysoczańska et al. (2023).

| Method | ❄ Params | 🔥 Params | Pascal VOC | Pascal Context | COCO-Stuff | Cityscapes | ADE20K |
|---|---|---|---|---|---|---|---|
| Miscellaneous | | | | | | | |
| ReCo (Shin et al., 2022) | 313M | 0 | 57.7 | 22.3 | 14.8 | 21.1 | 11.2 |
| GroupViT (Xu et al., 2022) | 0 | 55M | 79.7 | 23.4 | 15.3 | 11.1 | 9.2 |
| TCL (Cha et al., 2023) | 156M | 21M | 77.5 | 30.3 | 19.6 | 23.1 | 14.9 |
| MaskCLIP (Dong et al., 2023) | 291M | 0 | 74.9 | 26.4 | 16.4 | 12.6 | 9.8 |
| OVDiff (Karazija et al., 2023) | 1,226M | 0 | 81.7 | 33.7 | - | - | 14.9 |
| CLIP-DINOiser (Wysoczańska et al., 2023) | - | - | 80.9 | 35.9 | 24.6 | 31.7 | 20.0 |
| LAION-400M | | | | | | | |
| CLIP (Radford et al., 2021) (ViT-B) | 0 | 157M | 35.1 | 7.7 | 4.2 | 1.8 | 2.0 |
| LiT† (Zhai et al., 2022) (ViT-B) | 94M | 63M | 80.5 | 31.8 | 23.3 | 24.7 | 18.7 |
| SimZSS (ViT-B) | 94M | 63M | 85.1 | 34.2 | 24.9 | 27.8 | 19.6 |
| COCO Captions | | | | | | | |
| LiT† (Zhai et al., 2022) (ViT-B) | 94M | 63M | 86.1 | 35.5 | 25.6 | 25.8 | 18.1 |
| SimZSS (ViT-S) | 21M | 40M | 87.2 | 37.3 | 23.8 | 29.2 | 17.9 |
| SimZSS (ViT-B) | 94M | 63M | **90.3** | **43.1** | **29.0** | **33.0** | **21.8** |

We validate the proposed method on a pixel-level zero-shot segmentation task. In this scenario, the model relies solely on textual class descriptions to classify image pixels. However, accurately representing the `background` class can be challenging due to dataset-specific properties not fully captured by text descriptions. As such, we follow previous works (Wysoczańska et al., 2023; Cha et al., 2023) and first evaluate without the `background` class. We follow the MMSegmentation (Contributors, 2020) implementation of Cha et al. (2023). Specifically, images are resized to have a shorter side of 448 pixels, and inference is performed with a sliding window of $448 \times 448$ and a stride of 224 pixels. ImageNet templates from Radford et al. (2021) are used to contextualize each class name before obtaining their textual representation. Finally, the predictions obtained by projecting patch representations onto class names, are up-sampled using bilinear interpolation to restore the predictions to the original image size. We report the mIoU scores across five standard datasets, namely Pascal VOC (Everingham et al., 2012), Pascal Context (Mottaghi et al., 2014), COCO-Stuff (Caesar et al., 2018), Cityscapes (Cordts et al., 2016) and ADE20K (Zhou et al., 2017). For datasets that encompass the `background` class, the corresponding annotations and pixels are simply ignored. For the sake of clarity, we report the results without any post-processing techniques such as Pixel-Adaptive Mask Refinement (PAMR) (Araslanov & Roth, 2020).

In Table 1, we observe that our method using a ViT-S/14 as the vision tower is already competitive and that shifting to a larger vision backbone, namely a ViT-B/14, yields state-of-the-art results on all datasets when trained on COCO Captions. In particular, it can be seen that the proposed pipeline significantly outperforms models pretrained with LiT (Zhai et al., 2022) under the same conditions. This suggests that the reported performance is not solely attributable to the freezing of the vision tower or DINOv2 (Oquab et al., 2023) visual pretraining. The results obtained with LAION-400M indicate that SimZSS is compatible with large-scale uncurated datasets, exhibiting overall strong performance and improvements compared to models trained with LiT in the same setting. However, it is also apparent that both LiT and SimZSS seem to benefit more from curation than scale for segmentation. This trend contrasts with findings in classification tasks, as shown in Table 3.

Finally, we present the performance of a CLIP-based model trained with a ViT-B/16 vision backbone on the LAION-400M dataset for 32 epochs (Ilharco et al., 2021). The results highlight that, while CLIP excels at image-level tasks such as zero-shot classification, it demonstrates limitations

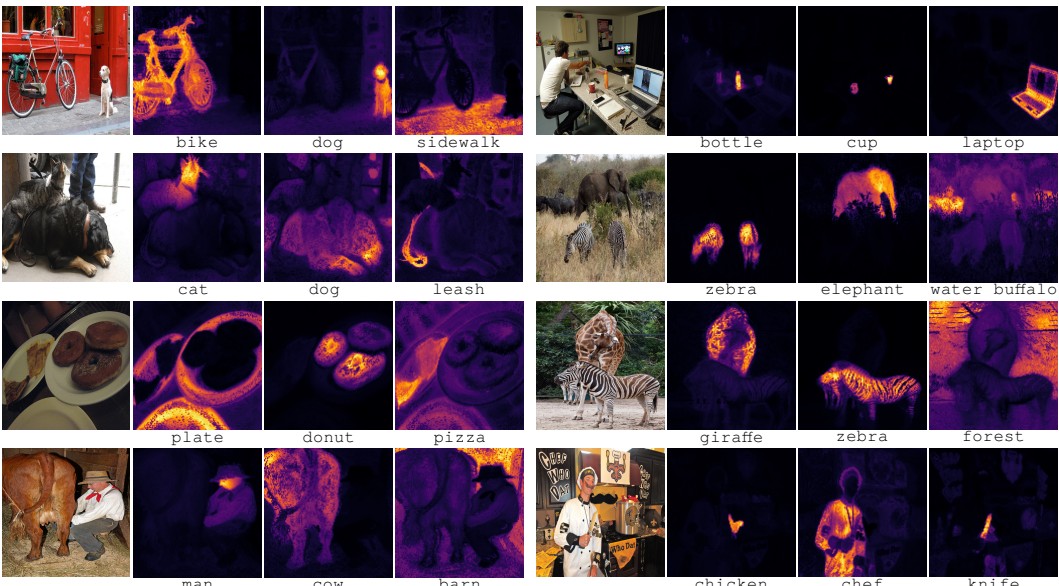

Figure 3: **Vision-language alignment of text concepts and dense visual representations.** Concepts present in the image are embedded independently by the text encoder and then projected onto the representations of each patch within the image. The images are processed at a resolution of $896 \times 896$ pixels, corresponding to $4\times$ the training resolution. The alignment is performed on LAION-400M using a ViT-B/14 as the vision tower.

Table 2: **Zero-shot segmentation.** Pixel-wise predictions of the foreground classes are obtained by projecting patch representations onto pre-computed text embeddings of the class names, followed by up-sampling. Predictions that fall below a predetermined threshold are assigned to the background class. The mIoU scores are reported across three standard segmentation datasets. † refers to our reproduction using DINOv2 pretrained vision backbones. The remaining results are as reported in Wysoczańska et al. (2023).

| Method | ❄ Params | 🔥 Params | Pascal Context | COCO-Object | Pascal VOC |
|---|---|---|---|---|---|
| *Miscellaneous* | | | | | |
| ReCo (Shin et al., 2022) | 313M | 0 | 19.9 | 15.7 | 25.1 |
| OVDiff (Karazija et al., 2023) | 1,226M | 0 | 30.1 | 34.8 | **67.1** |
| GroupViT (Xu et al., 2022) | 0 | 55M | 18.7 | 27.5 | 50.4 |
| ZeroSeg (Chen et al., 2023) | - | - | 21.8 | 22.1 | 42.9 |
| SegCLIP (Luo et al., 2023) | - | - | 24.7 | 26.5 | 52.6 |
| TCL (Cha et al., 2023) | 156M | 21M | 24.3 | 30.4 | 51.2 |
| CLIPpy (Ranasinghe et al., 2023) | - | - | - | 32.0 | 52.2 |
| OVSegmentor (Xu et al., 2023a) | - | - | 20.4 | 25.1 | 53.8 |
| CLIP-DIY (Wysoczańska et al., 2024) | - | - | 19.7 | 31.0 | 59.9 |
| MaskCLIP (Dong et al., 2023) | 291M | 0 | 23.6 | 20.6 | 38.8 |
| CLIP-DINOiser (Wysoczańska et al., 2023) | - | - | 32.4 | 34.8 | 62.1 |
| *LAION-400M* | | | | | |
| CLIP (Radford et al., 2021) (ViT-B) | 0 | 157M | 6.9 | 5.2 | 11.1 |
| LiT† (Zhai et al., 2022) (ViT-B) | 94M | 63M | 29.6 | 38.3 | 48.1 |
| SimZSS (ViT-B) | 94M | 63M | 31.1 | 38.1 | 48.6 |
| *COCO Captions* | | | | | |
| LiT† (Zhai et al., 2022) (ViT-B) | 94M | 63M | 31.5 | 39.5 | 51.4 |
| SimZSS (ViT-S) | 23M | 40M | 32.8 | 39.5 | 55.5 |
| SimZSS (ViT-B) | 94M | 63M | **37.2** | **43.5** | 58.4 |

in achieving fine-grained cross-modal alignment. In Appendix A.3, we address this limitation and provide evidence suggesting that it is not solely attributable to vision-language misalignment.

## 4.3 ZERO-SHOT SEGMENTATION

In a second scenario, we explore a zero-shot segmentation task including `background` class. Similar to Cha et al. (2023), we do not rely on the textual representation to predict the `background`

Table 3: **Zero-shot classification.** Image-level predictions are obtained by projecting the image `[CLS]` token onto pre-computed text embeddings of class names. Accuracy is reported for various visual pretraining and vision-language alignment methods. † refers to our reproduction using DINOv2 pretrained vision backbones. The remaining results are as reported in Zhai et al. (2022).

| Method | Visual pretraining | Backbone | Pretraining dataset | Alignment dataset | Alignement samples | Labels | ImageNet-1k | Average |
|---|---|---|---|---|---|---|---|---|
| LiT (Zhai et al., 2022) | MoCo-v3 (Chen et al., 2021) | ViT-B/16 | ImageNet-1k | CC12M+YFCC100M | - | ✗ | 55.4 | - |
| LiT (Zhai et al., 2022) | DINOv1 (Caron et al., 2021) | ViT-B/16 | ImageNet-1k | CC12M+YFCC100M | - | ✗ | 55.5 | - |
| LiT (Zhai et al., 2022) | AugReg (Steiner et al., 2022) | ViT-B/16 | ImageNet-21k | CC12M+YFCC100M | - | ✓ | 55.9 | - |
| LiT† (Zhai et al., 2022) | DINOv2 (Oquab et al., 2023) | ViT-B/14 | LVD-142M | COCO Captions | 4M | ✗ | 22.6 | 24.4 |
| LiT† (Zhai et al., 2022) | DINOv2 (Oquab et al., 2023) | ViT-B/14 | LVD-142M | LAION-400M | 400M | ✗ | 63.6 | 37.5 |
| CLIP (Radford et al., 2021) | - | ViT-B/16 | - | LAION-400M | 12.8 B | ✗ | 67.0 | **47.2** |
| *Ours* | | | | | | | | |
| SimZSS | DINOv2 (Oquab et al., 2023) | ViT-B/14 | LVD-142M | COCO Captions | 4M | ✗ | 24.3 | 26.1 |
| SimZSS | DINOv2 (Oquab et al., 2023) | ViT-B/14 | LVD-142M | LAION-400M | 400M | ✗ | 64.1 | 38.9 |
| SimZSS | DINOv2 (Oquab et al., 2023) | ViT-B/14 | LVD-142M | LAION-400M | 1.6B | ✗ | **69.3** | 41.3 |

class, but rather on the confidence levels of the predictions on the remaining classes. More precisely, we assign a given pixel to the `background` class if the highest confidence score among the other class predictions falls below a dataset and model-specific threshold. The remaining implementation details are identical to the above-described setting. The mIoU scores are reported on three datasets, namely Pascal Context (Mottaghi et al., 2014), COCO-Object (Caesar et al., 2018) and Pascal VOC (Everingham et al., 2012).

In Table 2, we observe similar trends as in the scenario without the `background` class (see Tab. 1). Reported results are not as unequivocal as in the former evaluation. This is in part due to the crudeness of the `background` detection mechanism, which contrasts with the more sophisticated approaches used by some of the competing baselines. For instance, CLIP-DINOiser (Wysoczańska et al., 2023) relies on FOUND (Siméoni et al., 2023), whereas OVDiff (Karazija et al., 2023) uses different background prototypes for each class. Once again, training on COCO Captions leads to improved performance. When training on LAION-400M, a concept from the bank of concepts is identified in less than 15% of the samples on average. As a result, concepts rarely co-occur in a single sample, making it sufficient to detect the concept without localizing it in the image to minimize the loss. Therefore, the model is not trained to be less confident in the background.

## 4.4 ZERO-SHOT CLASSIFICATION

The proposed method derives from LiT (Zhai et al., 2022), noted for its excellent zero-shot classification performance on ImageNet-1k (Deng et al., 2009) among other benchmarks. Although our approach is tailored for dense downstream tasks, we investigate whether SimZSS exhibits zero-shot classification capabilities or if the introduced local consistency objective breaks this property. We follow the evaluation protocol from OpenCLIP (Ilharco et al., 2021). Class names are contextualized with the corresponding templates from CLIP (Radford et al., 2021) and embedded through the text encoder. On the vision side, images are passed through the vision encoder, and the global representation is used to obtain class predictions via projection onto the class embeddings. We evaluate on 38 datasets and report the average top-1 accuracy in Table 3. The accuracy on ImageNet-1k is also reported specifically. The complete zero-shot classification results on the 38 datasets are depicted in Figure 5.

The results reported in Table 3 reveal that SimZSS not only preserves classification capabilities comparable to LiT but also exhibits slight improvements in the low-data regime, namely when training on COCO Captions. It is worth mentioning that the performance of CLIP after 1 epoch is approximately $43\%$ (as depicted here). Overall, while pretraining on COCO Captions excels for

Table 4: **Training without the bank of concepts.** The impact of the concept bank is evaluated by training SimZSS on COCO Captions without it, that is, the concept-level objective is enforced for all noun phrases in the captions. The training is conducted with a ViT-B/14 pretrained with DINOv2. † indicates results reproduced using DINOv2 pretrained vision backbones.

| Method | Bank | Without background | | | | | With background | | |
|---|---|---|---|---|---|---|---|---|---|
| | | Pascal VOC | Pascal Context | COCO-Stuff | Cityscapes | ADE20K | Pascal Context | COCO-Object | Pascal VOC |
| LiT† (Zhai et al., 2022) | - | 86.1 | 35.5 | 25.6 | 25.8 | 18.1 | 31.5 | 39.5 | 51.4 |
| SimZSS | ✗ | 89.2 | 41.1 | 27.6 | 31.2 | 20.0 | 35.2 | 42.6 | 56.8 |
| SimZSS | ✓ | 90.3 | 43.1 | 29.0 | 33.0 | 21.8 | 37.2 | 43.5 | 58.4 |

segmentation tasks, zero-shot classification benefits from larger datasets. This suggests that a pre-training on LAION-400M followed by finetuning on COCO Captions could optimize the benefits of both datasets.

## 4.5 BANK OF CONCEPTS

At its core, SimZSS relies on identifying noun phrases within captions and filtering them through a bank of concepts. This filtering ensures that the identified noun phrases are valid candidates for visual concepts and can be used to query the images. A key question that arises is whether the segmentation improvements of the proposed method are confined to the classes within the concept bank or if they generalize to other classes as well. To test this hypothesis, we trained SimZSS on COCO Captions without the concept bank.

As expected, the scores reported in Table 4 are slightly lower compared to using the concept bank. However, the results consistently outperform the baseline LiT, which focuses solely on enforcing consistency between caption and image representations. Moreover, we observe that the performance of SimZSS without the concept bank is closer to the version with the concept bank than to LiT. This demonstrates that SimZSS's performance is driven more by the consistency objective between noun phrases and localized image regions than by the concept bank itself. Considering the results achieved by concurrent baselines on the same tasks (see Tabs. 1 and 2), we conclude that SimZSS yields great performance compared to existing methods, irrespective of whether a concept bank is used, which shows that a concept does not need to appear in the concept bank to be properly localized at test time.

Alternatively, Figure 1 suggests that the vision tower, here a ViT-B/14 pretrained with DI-NOv2 (Oquab et al., 2023), could handle more concepts than those present in the concept bank. Indeed, some visual concepts, such as `bike` (third row, columns 1-5), appear well-segmented in the image but are not highlighted in the caption, meaning the concept is absent from the concept bank, and thus, no constraint is enforced. This claim is further supported by the fine-grained vision-language alignment depicted in Figure 3. The visualization further shows that, even though concepts such as `forest`, `chef`, `water buffalo`, or `bike` are not in the concept bank, the text concepts and the corresponding patches appear well aligned.

## 5 CONCLUSION

We introduce SimZSS, a simple framework to endow pretrained pure vision models with open-vocabulary segmentation capabilities. This approach is versatile: it can accommodate various back-bone sizes, pretraining methods, and datasets, irrespective of their scale and degree of curation. Overall, SimZSS is both a computationally and data-efficient method that yields excellent results across standard zero-shot segmentation benchmarks while preserving efficiency at inference time.

**Reproducibility Statement**  We will release our code and pretrained models to the public upon acceptance. The release will include a README file with detailed instructions for setting up the environment and running the training. All datasets used in our work are publicly available.

**Acknowledgement**  This project is funded by the Personalized Health and Related Technologies (PHRT), grant number 2021/344, as well as the European Research Council (ERC) under the European Union's Horizon 2020 research and innovation program (Grant Agreement No. 101021347). This work is also partially supported by funding from the Flemish Government under the "Onder-zoeksprogramma Artificiele Intelligentie (AI) Vlaanderen". This work was supported by a grant from the Swiss National Supercomputing Centre (CSCS) on the Swiss share of the LUMI system under project ID 606.

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

APPENDIX

# A    ADDITIONAL EXPERIMENTS

## A.1    IMPACT OF THE VISION TOWER

The underlying assumption of SimZSS is that the visual representations do not require further training, and only the vision-language alignment must be learned. As such, the quality of the vision backbone is pivotal to the overall performance of SimZSS.

In Tables 5 and 6, we explore the performance of various pretrained vision towers in zero-shot segmentation tasks. Our observations indicate that SimZSS is versatile and can integrate diverse models and pretraining approaches effectively. As expected, models with superior performance on pure vision tasks also show enhanced results when used with our method, as well as with LiT. Most importantly, regardless of the vision backbone employed, SimZSS consistently surpasses LiT. Additionally, employing denoised ViTs (Yang et al., 2024), which exhibit better semantic feature correlation, leads to improved segmentation. This outcome is anticipated since our approach assumes that features corresponding to semantically related regions are highly correlated (see Sec. 3.2).

As shown in Tables 5 and 6, using a ViT-L/14 as the vision backbone does not improve overall performance compared to the ViT-B/14, whether for LiT (Zhai et al., 2022) or SimZSS. This outcome cannot be attributed to overfitting due to the increased number of trainable parameters in the text encoder. Replacing the text encoder with the architecture tailored for the ViT-B/14 did not result in any performance gains. The results for the ViT-L/14 in Tables 5 and 6 were obtained using average pooling of the dense visual representation. This approach was necessary because we observed that the `[CLS]` token in this model was poorly aligned with the remaining tokens, making the training of both LiTand SimZSS unsuccessful. These findings align with the lower results reported by Pariza et al. (2024a) when using frozen features from DINOv2's ViT-L/14 for dense downstream tasks, as well as with the growing body of literature on the denoising of DINOv2 models (Yang et al., 2024; Wang et al., 2024).

## A.2    COMPARISON WITH TRAINING-FREE APPROACHES

We compare SimZSS with training-free approaches on the zero-shot segmentation benchmark. The results in Table 7 show that these methods, particularly ProxyCLIP (Lan et al., 2024), can achieve strong performance without requiring any training. Moreover, these methods retain CLIP's zero-shot classification capabilities.

While SimZSS does require training, the cost is manageable as it only takes 15 minutes on 8 GPUs (for training on COCO Captions) and offers several advantages. SimZSS enables the use of smaller models, such as a ViT-S/14, can be tailored to specific domains, and is generally less computationally intensive at inference time, which can effectively amortize the training cost.

Table 5: **Ablation study on the vision backbone for zero-shot foreground segmentation.** We investigate the effect of the vision backbone's size and pretraining method on the downstream task. The impact of using artifact-free models is also evaluated (with/without DVT (Yang et al., 2024)). We report the mIoU scores across five standard datasets. † refers to our reproduction using DINOv2 pretrained vision backbones.

| Method | Pretraining | Backbone | DVT | Pascal VOC | Pascal Context | COCO-Stuff | Cityscapes | ADE20K |
|---|---|---|---|---|---|---|---|---|
| *Supervised* | | | | | | | | |
| LiT† (Zhai et al., 2022) | AugReg | ViT-B/16 | ✗ | 74.1 | 18.3 | 13.3 | 15.6 | 10.2 |
| SimZSS | AugReg | ViT-B/16 | ✗ | **81.2** | **23.3** | **16.1** | **18.9** | **12.6** |
| *Self-supervised* | | | | | | | | |
| LiT† (Zhai et al., 2022) | DINOv2 | ViT-S/14 | ✗ | 84.0 | 32.8 | 22.0 | 24.0 | 16.3 |
| SimZSS | DINOv2 | ViT-S/14 | ✗ | 87.2 | 37.3 | 23.8 | 29.2 | 17.9 |
| LiT† (Zhai et al., 2022) | DINOv2 | ViT-B/14 | ✗ | 86.3 | 34.1 | 24.6 | 25.8 | 17.1 |
| SimZSS | DINOv2 | ViT-B/14 | ✗ | 87.4 | 41.1 | 26.6 | 31.6 | 19.9 |
| LiT† (Zhai et al., 2022) | DINOv2 | ViT-B/14 | ✓ | 86.1 | 35.5 | 25.6 | 25.8 | 18.1 |
| SimZSS | DINOv2 | ViT-B/14 | ✓ | **90.3** | **43.1** | **29.0** | **33.0** | **21.8** |
| LiT† (Zhai et al., 2022) | DINOv2 | ViT-L/14 | ✗ | 81.5 | 33.5 | 23.1 | 26.0 | 17.6 |
| SimZSS | DINOv2 | ViT-L/14 | ✗ | 83.9 | 36.4 | 24.5 | 31.1 | 18.8 |

Table 6: **Ablation study on the vision backbone for zero-shot segmentation.** We investigate the effect of the vision backbone's size, pretraining method, and patch size on the downstream task. The impact of using artifact-free models is also evaluated (with/without DVT (Yang et al., 2024)). We report the mIoU scores on three standard datasets.

| Method | Pretraining | Backbone | DVT | Pascal Context | COCO-Object | Pascal VOC |
|---|---|---|---|---|---|---|
| *Supervised* | | | | | | |
| LiT (Zhai et al., 2022) | AugReg | ViT-B/16 | ✗ | 17.8 | 26.1 | 32.2 |
| SimZSS | AugReg | ViT-B/16 | ✗ | **21.6** | **29.7** | **35.9** |
| *Self-supervised* | | | | | | |
| LiT (Zhai et al., 2022) | DINOv2 | ViT-S/14 | ✗ | 29.9 | 36.5 | 49.6 |
| SimZSS | DINOv2 | ViT-S/14 | ✗ | 32.8 | 39.5 | 55.5 |
| LiT (Zhai et al., 2022) | DINOv2 | ViT-B/14 | ✗ | 30.5 | 28.7 | 49.8 |
| SimZSS | DINOv2 | ViT-B/14 | ✗ | 35.7 | 40.5 | 57.5 |
| LiT (Zhai et al., 2022) | DINOv2 | ViT-B/14 | ✓ | 31.5 | 39.5 | 51.4 |
| SimZSS | DINOv2 | ViT-B/14 | ✓ | **37.2** | **43.5** | **58.4** |
| LiT (Zhai et al., 2022) | DINOv2 | ViT-L/14 | ✗ | 29.2 | 33.2 | 42.1 |
| SimZSS | DINOv2 | ViT-L/14 | ✗ | 31.9 | 34.6 | 46.6 |

Table 7: **Comparison with training-free approaches.** Training-free approaches are compared to SimZSS trained on COCO Captions with different vision backbones pre-trained with DINOv2. Results for the training-free approaches are reported as in Lan et al. (2024).

| Method | Backbone | Without background | | | | | With background | | |
| | | Pascal VOC | Pascal Context | COCO-Stuff | Cityscapes | ADE20K | Pascal Context | COCO-Object | Pascal VOC |
|---|---|---|---|---|---|---|---|---|---|
| CLIPSurgery (Li et al., 2023c) | ViT-Base | - | - | 21.9 | 31.4 | - | 29.3 | - | - |
| GEM (Bousselham et al., 2024) | ViT-Base | - | - | - | - | 15.7 | 32.6 | - | 46.2 |
| MaskCLIP (Zhou et al., 2022a) | ViT-Base | 74.9 | 26.4 | 16.4 | 12.6 | 9.8 | 23.6 | 20.6 | 38.8 |
| SCLIP (Wang et al., 2025) | ViT-Base | 80.4 | 34.2 | 22.4 | 32.2 | 16.1 | 30.4 | 30.5 | 59.1 |
| ProxyCLIP (Lan et al., 2024) | ViT-Base | 80.3 | 39.1 | 26.5 | 38.1 | 20.2 | 35.3 | 37.5 | **61.3** |
| MaskCLIP (Zhou et al., 2022a) | ViT-Large | 29.4 | 12.4 | 8.8 | 11.5 | 7.2 | 11.7 | 7.2 | 23.3 |
| SCLIP (Wang et al., 2025) | ViT-Large | 69.1 | 22.3 | 17.6 | 18.6 | 10.9 | 22.3 | 25.0 | 43.5 |
| ProxyCLIP (Lan et al., 2024) | ViT-Large | 83.1 | 37.7 | 25.6 | **40.1** | **22.6** | 34.5 | 39.2 | 60.6 |
| SimZSS | ViT-Small | 87.2 | 37.3 | 23.8 | 29.2 | 17.9 | 32.8 | 39.5 | 55.5 |
| SimZSS | ViT-Base | **90.3** | **43.1** | **29.0** | 33.0 | 21.8 | **37.2** | **43.5** | 58.4 |
| SimZSS | ViT-Large | 83.9 | 36.4 | 24.5 | 31.1 | 18.8 | 31.9 | 34.6 | 46.6 |

## A.3 IDENTIFYING BOTTLENECKS

To pinpoint potential areas for improvement, we take a close look at the performance of SimZSS in the context of zero-shot segmentation, directly comparing it to other segmentation methods on the same datasets and backbone. To dissect the influence of vision-language alignment on overall performance, we strip out the language model's class name embeddings and replace them with visual embeddings. These visual embeddings, derived from the training set, are computed by average pooling of the corresponding local representations for each class.

In Table 8, we report the results of the above-described experiment using the ViT-B/14 from DINOv2. For datasets that include a `background` class, we explore two strategies: *i)* treating the background as a distinct class with its own concept, and *ii)* classifying a given patch or pixel as background when the confidence of the prediction across all foreground classes is low.

The results support the hypothesis that the background is, in general, too heterogeneous to be well captured by a single concept, even in the absence of a modality gap. We observe that the few-shots strategy generally outperforms the zero-shot one. The high variability of the difference in mIoU scores between two strategies across different datasets indicates that the modality gap is not the sole reason for the observed discrepancies. In the context of zero-shot segmentation, the concepts are common to all datasets, while in few-shot segmentation, they are dataset-specific. This is well-reflected by the similarity of the two strategies on COCO-Stuff and COCO-Object, which can be attributed to the alignment of the pretraining dataset (COCO Captions) with the downstream datasets. Overall, this suggests that the image-caption pairs dataset is a stronger limiting factor than the modality gap itself. Leveraging image-captioning methods (Li et al., 2023a; Liu et al., 2024) to generate captions for target datasets could prove beneficial. The performance achieved through dense nearest-neighbor retrieval or linear segmentation using the same vision backbone on the same datasets indicates that there is still plenty of room for improvement.

Conversely, Table 10 from DINOv2 (Oquab et al., 2023) demonstrates significantly superior performance on the linear segmentation downstream evaluation across all their models, including ViT-

Table 8: **Few-shot vs. zero-shot segmentation comparison**. We compare the zero-shot segmentation performance of SimZSS with an alternative approach that constructs a bank of visual concepts from the training set rather than relying on embedded class names (*i.e.*, textual concepts). For reference, we also include the performance of linear segmentation and dense nearest-neighbor retrieval, as reported in Pariza et al. (2024b).

| Method | Background | Without background | | | | | With background | | |
|---|---|---|---|---|---|---|---|---|---|
| | | Pascal VOC | Pascal Context | COCO-Stuff | Cityscapes | ADE20K | Pascal Context | COCO-Object | Pascal VOC |
| Dense k-NN | Concept | - | - | - | - | 38.7 | - | - | 76.2 |
| Linear | Concept | - | - | 57.6 | - | 40.3 | - | 78.3 | 79.8 |
| Few-shots | Concept | 91.4 | 51.0 | 31.3 | 49.7 | 31.7 | 43.7 | 27.7 | 51.6 |
| Few-shots | Confidence | 91.4 | 51.0 | 31.3 | 49.7 | 31.7 | 43.6 | 42.4 | 62.6 |
| SimZSS | Confidence | 90.3 | 43.1 | 29.0 | 33.0 | 21.8 | 37.2 | 43.5 | 58.4 |

Table 9: **Ablation study on the impact of the loss weight $\lambda$ and the temperature parameter $\tau$**. Training is performed on COCO Captions using a ViT-B/14 vision backbone. The performance is evaluated through a comparison of mIoU scores on the zero-shot segmentation task.

| Loss weight | Temperature | Without background | | | | | With background | | |
|---|---|---|---|---|---|---|---|---|---|
| | | Pascal VOC | Pascal Context | COCO-Stuff | Cityscapes | ADE20K | Pascal Context | COCO-Object | Pascal VOC |
| $\lambda = 0.00$ | - | 86.1 | 35.5 | 25.6 | 25.8 | 18.1 | 31.5 | 39.5 | 51.4 |
| $\lambda = 0.01$ | $\tau = 0.1$ | 89.8 | 40.2 | 27.6 | 29.7 | 19.6 | 34.9 | 42.1 | 54.9 |
| $\lambda = 0.02$ | $\tau = 0.1$ | 89.8 | 41.9 | 28.2 | 30.9 | 20.4 | 36.1 | 42.3 | 57.1 |
| $\lambda = 0.05$ | $\tau = 0.1$ | **90.3** | **43.1** | **29.0** | 33.0 | **21.8** | **37.2** | **43.5** | 58.4 |
| $\lambda = 0.10$ | $\tau = 0.1$ | 90.1 | 41.8 | 28.6 | **33.6** | 22.1 | 36.1 | 43.1 | **58.5** |
| $\lambda = 0.05$ | $\tau = 0.01$ | 89.2 | 42.3 | 28.8 | 33.1 | 21.6 | 36.6 | 43.2 | 57.0 |
| $\lambda = 0.05$ | $\tau = 0.04$ | 89.7 | 42.7 | 28.7 | **34.3** | **22.1** | 36.7 | 42.9 | 57.0 |
| $\lambda = 0.05$ | $\tau = 0.07$ | 90.0 | 42.9 | 28.9 | 33.4 | 21.3 | 37.0 | 43.1 | 58.8 |
| $\lambda = 0.05$ | $\tau = 0.10$ | **90.3** | **43.1** | **29.0** | 33.0 | 21.8 | **37.2** | **43.5** | **58.4** |
| $\lambda = 0.05$ | $\tau = 0.40$ | 88.7 | 40.0 | 27.6 | 28.2 | 19.5 | 34.4 | 42.1 | 52.6 |

S/14, compared to OpenCLIP ViT-G/14 (Ilharco et al., 2021). This underscores the limitations of the cross-modal contrastive objective in learning effective visual representations. Moreover, it suggests that methods relying on pretrained CLIP-based models face a lower performance ceiling compared to our approach. Indeed, zero-shot segmentation can be viewed as pixel- or patch-level classification using a frozen linear layer initialized with the embeddings of textual labels.

## A.4 SimZSS-specific hyperparameters

Thanks to its simplicity, SimZSS has few hyperparameters, with the primary ones being $\tau$, which regulates the temperature used to query the dense visual representation, and $\lambda$, which modulates the contribution of the dense loss to the overall objective (see Eqs. (4) and (10)). After a grid search on $\lambda$, $\tau$, and the learning rate, we find that the best-performing setting for training on COCO Captions is $(\lambda = 0.05, \tau = 0.1)$. In Table 9, we report the resulting zero-shot segmentation performance when varying $\lambda$ and $\tau$ around this optimal point.

It can be observed that the setting with $\lambda = 0$, corresponding to LiT (Zhai et al., 2022), performs the worst. Regarding $\tau$, SimZSS works best with low temperatures. This is not surprising, as in this setting, fewer patches contribute to the representation of the visual concepts, aligning more closely with the downstream evaluations.

## A.5 Scalability of SimZSS

An important property of vision-language pretraining methods is their capacity to leverage large datasets effectively, which necessitates them to be *i)* compatible with noisy image-caption pairs, and *ii)* computationally efficient.

We first verify in Tables 12 and 13 that SimZSS-specific operations incur only moderate memory overhead and negligible time overhead compared to LiT. Considering this, along with the data efficiency of the proposed approach, SimZSS is orders of magnitude cheaper to train than CLIP. Secondly, we escalate the number of image-caption pairs from LAION-400M during training and report the performance on zero-shot segmentation at the end of each training phase. The results depicted in Figure 4 underscore the scalability of our method SimZSS and suggest potential gains from even larger datasets, such as LAION-5B (Schuhmann et al., 2022), or increased training epochs.

Table 10: **Ablation study on the impact of the training resolution.** Training is conducted on COCO Captions with an image resolution of either $224 \times 224$ or $448 \times 448$ pixels using a ViT-B/14 vision backbone pretrained with DINOv2. Performance on the zero-shot segmentation task is then compared based on achieved mIoU scores. † refers to our reproduction using DINOv2 pretrained vision backbones.

| Method | Resolution | Without background | | | | | With background | | |
|---|---|---|---|---|---|---|---|---|---|
| | | Pascal VOC | Pascal Context | COCO-Stuff | Cityscapes | ADE20K | Pascal Context | COCO-Object | Pascal VOC |
| LiT† (Zhai et al., 2022) | $448 \times 448$ | 84.2 | 33.9 | 25.5 | 24.9 | 17.3 | 31.1 | 39.7 | 50.5 |
| SimZSS | $448 \times 448$ | 87.2 | 40.2 | 27.9 | 29.4 | 20.4 | 35.2 | 43.2 | 56.6 |
| LiT† (Zhai et al., 2022) | $224 \times 224$ | 86.1 | 35.5 | 25.6 | 25.8 | 18.1 | 31.5 | 39.5 | 51.4 |
| SimZSS | $224 \times 224$ | **90.3** | **43.1** | **29.0** | **33.0** | **21.8** | **37.2** | **43.5** | **58.4** |

## A.6 IMAGE RESOLUTION

Segmentation methods typically benefit from higher image resolutions during training. To assess whether this holds for SimZSS, we conducted training on COCO Captions, increasing the image resolution from $224 \times 224$ pixels to higher resolution images, $448 \times 448$ pixels. For that experiment, the vision backbone utilized was a ViT-B/14 pretrained with DINOv2. Apart from the increased resolution, all other training procedures remain unchanged.

As shown in Table 10, neither LiT nor SimZSS exhibited any performance gains when trained with higher-resolution images. This can likely be attributed to the fact that higher resolutions reduce the receptive field of each patch, leading to poorer semantic alignment between local representations and the global image representation, typically captured by the `[CLS]` token. In LiT Zhai et al. (2022), this misalignment hampers zero-shot segmentation performance, as it weakens the alignment between local visual features and the global textual representation, which is fundamental to the task. The same trend is observed with SimZSS, which builds upon the knowledge acquired from the alignment of global textual and visual representations.

We further explore the effect of using higher resolutions during evaluation. After training with an input resolution of $224 \times 224$, we assess zero-shot segmentation performance using a target size for the shorter side of either 448 or 896. As shown in Table 11, increasing the evaluation resolution proves beneficial primarily for the Cityscapes (Cordts et al., 2016) dataset. This is expected, given that Cityscapes consists of high-resolution images that undergo substantial downscaling, leading to a greater potential loss of information.

## A.7 HIGH-LEVEL PROFILING & RUNTIME COMPARISON

In Table 12, we present a profiling of the operations performed during a training step of SimZSS. Overall, SimZSS-specific operations account for less than $1\%$ of the total runtime. Table 13 further confirms that SimZSS runtime is comparable to that of LiT (Zhai et al., 2022) and adds modest memory overhead.

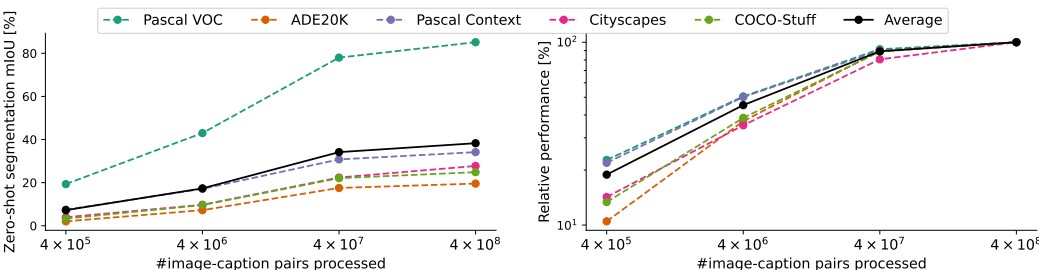

Figure 4: **Zero-shot segmentation performance as a function of the number of processed image-caption pairs in LAION-400M.** The left plot shows the mIoU percentages for different datasets, while the right plot shows the relative performance percentages. Each data point represents the result of running a vision-language alignment from scratch using SimZSS; these are not training curves.

Table 11: **Ablation study on the impact of the evaluation resolution.** Training is conducted on COCO Captions with an image resolution of $224 \times 224$ pixels using a ViT-B/14 vision backbone pretrained with DINOv2. Zero-shot segmentation performance is reported for images resized to a shorter side of either 448 or 896 pixels. † refers to our reproduction using DINOv2 pretrained vision backbones.

| Method | Shorter side | Without background | | | | | With background | | |
| | | Pascal VOC | Pascal Context | COCO-Stuff | Cityscapes | ADE20K | Pascal Context | COCO-Object | Pascal VOC |
|---|---|---|---|---|---|---|---|---|---|
| LiT† (Zhai et al., 2022) | 448 | 86.1 | 35.5 | 25.6 | 25.8 | 18.1 | 31.5 | 39.5 | 51.4 |
| SimZSS | 448 | **90.3** | 43.1 | **29.0** | 33.0 | 21.8 | 37.2 | **43.5** | 58.4 |
| LiT† (Zhai et al., 2022) | 896 | 79.7 | 34.0 | 23.8 | 30.4 | 20.6 | 31.0 | 37.2 | 49.2 |
| SimZSS | 896 | 86.1 | **43.3** | 28.9 | **36.1** | **22.7** | **37.7** | 41.7 | **59.9** |

Table 12: **High-level profiling of SimZSS**. We report the absolute and relative times of the various operations performed in SimZSS. Experiments are conducted on a single node with 4x AMD MI250x GPUs (2 compute dies per GPU, i.e., `worldsize = 8`) with a memory usage of 38GB per compute die. The backbone used is ViT-B/14, and the batch size is set to 1024 per compute die, totaling 8192. SimZSS-specific operations are highlighted.

| Description | Operation | Absolute time per iteration [ms] | Relative time [%] |
|---|---|---|---|
| Forward pass vision | $f_v(\cdot)$ | 1435.0 | 69.3 |
| Forward pass text | $f_t(\cdot)$ | 194.0 | 9.3 |
| Vision concept extraction | Equations (3) and (4) | 12.0 | 0.6 |
| Features gathering | - | 22.5 | 1.1 |
| Global consistency | Equation (6) | 1.0 | 0.05 |
| Concept-level consistency | Equation (9) | 0.9 | 0.04 |
| Weights update | `backpropagation` | 404.4 | 19.5 |
| *Total* | | 2069.8 | 100 |

# B   EVALUATION DATASETS

**Pascal VOC 2012**   The Pascal VOC dataset (Everingham et al., 2010) contains 20 classes with semantic segmentation annotations. The training set consists of 1,464 images, while the validation set includes 1,449 images. An additional `background` class is provided.

**Pascal Context**   The Pascal Context dataset (Mottaghi et al., 2014) extends the Pascal VOC dataset by providing detailed annotations for entire scenes. It includes 60 classes (including a `background` class) of semantic segmentation annotations. The training set contains approximately 4,998 images, while the validation set includes around 5,105 images.

**COCO-Stuff**   COCO-Stuff (Caesar et al., 2018) is an extension of COCO (Lin et al., 2014), providing pixel-wise annotations for 80 things classes and 91 stuff categories. The annotations are exhaustive, *i.e.*, no pixel remains unlabeled. It includes over 164K images for training and 20K images for validation, covering a wide range of scenes and objects.

**COCO-Object**   COCO-Object uses the same set of images as the above-described COCO-Stuff, but only contains labels for the "things" categories. An additional `background` label covers the remaining pixels.

**ADE20K**   The ADE20K dataset (Zhou et al., 2017) comprises a diverse set of images with annotations for 150 semantic categories. The training set includes 20,210 images, and the validation set consists of 2,000 images.

**Cityscapes**   The Cityscapes dataset (Cordts et al., 2016) contains high-resolution images of urban street scenes, annotated for 30 classes. The training set includes 2,975 images, and the validation set includes 500 images captured from 50 cities for semantic segmentation and urban scene understanding.

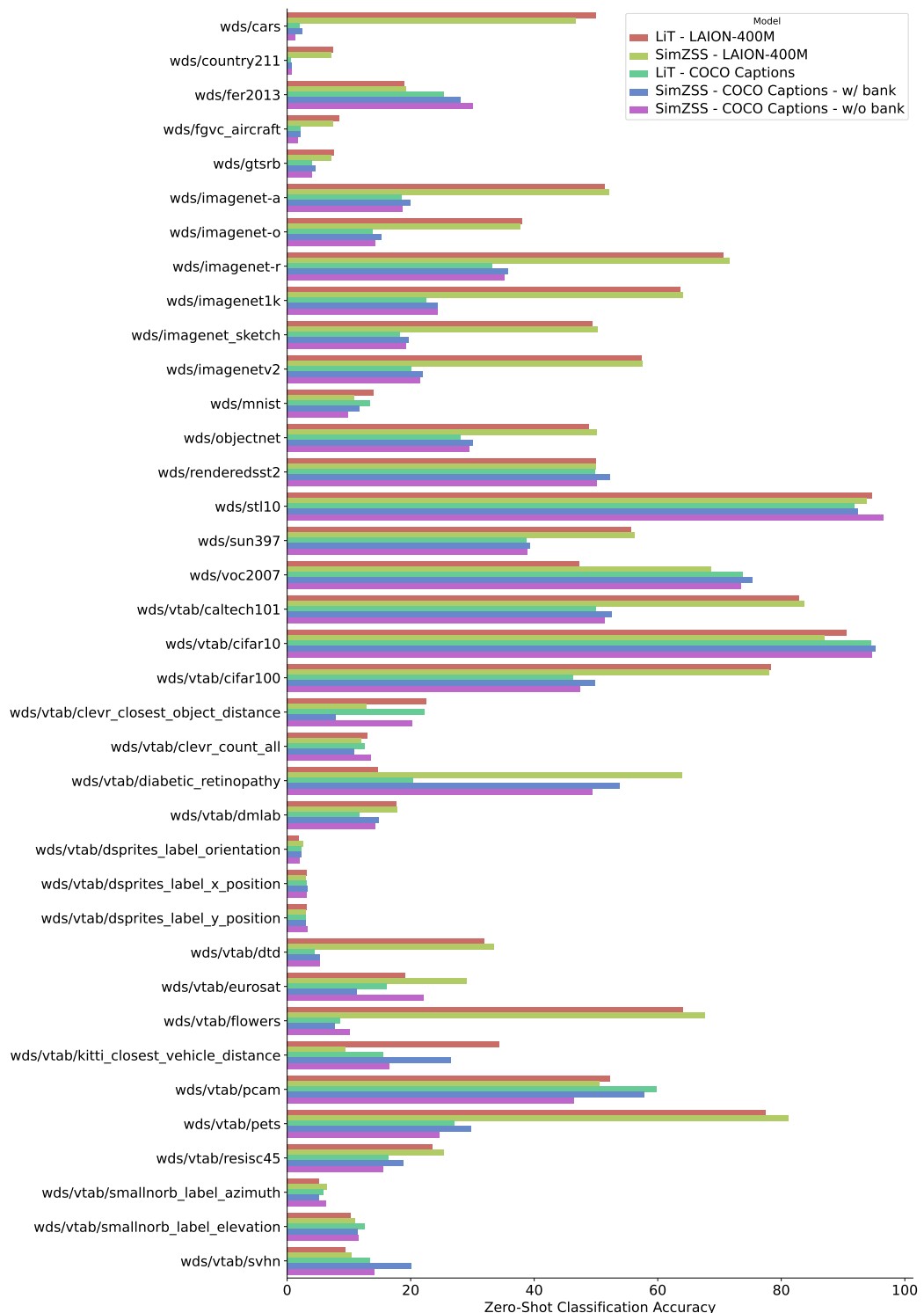

Figure 5: **Zero-shot classification benchmark.** We report the top-1 accuracy of SimZSS with and without the concept bank on 38 evaluation datasets when trained on COCO Captions and LAION-400M. Additional comparison with LiT is provided.

Table 13: **Computational and memory efficiency.** The efficiency of SimZSS is compared to that of related methods, *i.e.*, LiT and CLIP. When feasible, we report results using the local training batch size; otherwise, the largest power of 2 that fits into memory is utilized. The reported values are obtained on a single node equipped with 4x AMD MI250x (2 compute die per GPU, i.e., `worldsize = 8`).

| Method | Batchsize per compute die | Memory per compute die [GB] | Time per step [ms] | Throughput [img/s] |
|--------|---------------------------|------------------------------|---------------------|---------------------|
| CLIP   | 256                       | $\sim 40$                    | 1196.0              | 1712                |
| LiT    | 1024                      | $\sim 27$                    | 2049.2              | 3997                |
| SimZSS | 1024                      | $\sim 38$                    | 2069.8              | 3957                |

