# OpenReview forum: "A Simple Framework for Open-Vocabulary Zero-Shot Segmentation"
_ICLR.cc/2025/Conference — ICLR 2025 Poster_

### Official Review · Reviewer_enbH · 2024-10-29

**Soundness:** 3
**Presentation:** 3
**Contribution:** 3
**Rating:** 8
**Confidence:** 4

**Summary:**

The work proposed SimZSS, a simple and efficient framework for open-vocabulary zero-shot semantic segmentation. The work builds on top of a frozen pre-trained vision-only model and aligns a text encoder to achieve cross-modality concept-level alignment. Despite its simplicity, SimZSS achieves state-of-the-art results on seven out of eight benchmarks.

**Strengths:**

- The authors propose a simple yet effective approach for the task that exploits pre-trained vision models.
- The method is both data- and compute-efficient, as it does not require long training periods and needs captioned images for training.
- The method achieves state-of-the-art results on seven benchmark datasets.
- The method takes inspiration from LiT and extends it for the segmentation task while maintaining its classification capabilities.

**Weaknesses:**

- The work relies on a bank of pre-defined concepts to pre-process the captions and extract concepts to segment, which may represent a limitation in some situations. For example, the concept bank may not cover the diversity of potential objects in the image.
- While performance is competitive when using ViT-B, scaling the model backbone to a ViT-L model does not improve performance.

**Questions:**

- What are the current limitations of the concept bank used? Would a larger bank lead to better generalization? What are the complexities in scaling the concept bank to, e.g., WordNet or other taxonomies?
- As mining concepts from the captions provide promising results, I wonder how well the approach would work on more noisy captions, e.g., from web-scale datasets. How sensitive is SimZSS to such noise? Are there any techniques that could be employed to improve the robustness of the generation of the concept bank?

---

> ### Author Response · Authors · 2024-11-20
> **Reply part 1**
>
> ## Overview
> We thank the reviewer for the detailed and constructive feedback. Below, we address your questions and concerns in detail.
>
> ## Usage of a predefined concept bank & open-vocabulary capabilities
>
> There appears to be a general misunderstanding regarding the role of the concept bank. Its primary purpose is to filter out noun phrases that may not appear in the image, addressing noise in the captions of large-scale web datasets like LAION-400M. For more curated image-caption pair datasets, the concept bank is optional. To support this, we trained SimZSS without the concept bank (see Table 1 and 2 below). As expected, the scores are slightly lower compared to using the concept bank. However, the results consistently outperform the baseline (LiT), which focuses solely on enforcing consistency between captions and image representations. Moreover, we observe that the performance of SimZSS without the concept bank is closer to the version with the concept bank than to LiT. This demonstrates that SimZSS's performance is driven more by the consistency objective between noun phrases and localized image regions than by the concept bank itself. Considering the results achieved by concurrent baselines on the same tasks (Tables 1 and 2 in the main text), we conclude that SimZSS yields great performance compared to existing methods, irrespective of whether a concept bank is used, which shows that a concept does not need to appear in the concept bank to be properly localized at test time.
>
> #### Table 1: Zero-shot segmentation with and with out the bank of concepts on datasets without background.
> |  Method  |   Bank  |  Pascal VOC | Pascal Context | COCO-Stuff | Cityscapes | ADE20K |
> |:--------:|:-------:|:-----------:|:--------------:|:----------:|:----------:|:------:|
> | LiT    |  - | 86.1 | 35.5| 25.6|25.8|18.1|
> | SimZSS | w/o| 89.2 | 41.1| 27.6|31.2|20.0|
> | SimZSS | w/ | 90.3 | 43.1| 29.0|33.0|21.8|
>
>
> #### Table 2: Zero-shot segmentation with and with out the bank of concepts on datasets with background.
> |  Method  |   Bank  |  Pascal Context | COCO-Object | Pascal VOC |
> |:--------:|:-------:|:-----------:|:--------------:|:-----------:|
> |LiT   |  - |31.5|39.5|51.4|
> |SimZSS| w/o|35.2|42.6|56.8|
> |SimZSS| w/ |37.2|43.5|58.4|
>
>
> ## Performance with the ViT-L model
>
> We encountered significant issues with the ViT-L checkpoint from DINOv2, which arise from a misalignment between the CLS token and the spatial tokens. This is discussed in detail in lines 846–855 of the draft. Notably, LiT exhibits the same challenges with this specific model, leading us to conclude that the issue is inherent to the checkpoint itself rather than the methods being evaluated.
>
>
> ## Scalability of the bank of concepts and generalization
>
> The weights of the linear classifier are computed by summing the representations of the unique text concepts within a given batch. Consequently, the method's complexity depends solely on the number of unique concepts in a batch, rather than the total number of concepts in the entire bank. This ensures that scaling the concept bank does not introduce additional computational cost.
>
> To shed some light on the relation between generalization and the choice of the concept bank we evaluate SimZSS on a standard open-vocabulary zero-shot classification benchmark (see https://github.com/LAION-AI/CLIP_benchmark). In Table 3, we report the average top-1 accuracy obtained therein. We observe that using the bank of concepts does not decrease the average zero-shot classification performance, which shows that using a relatively small set of concepts does not negatively affect the generalization. Additionally, we observe that in average SimZSS does **not** compromise the zero-shot classification accuracy compared to a LiT pre-training with the same backbone (ViT-B/14 pre-trained with DINOv2) and dataset.
>
> #### Table 3: Average zero-shot classification top-1 accuracy.
> |   SimZSS - LAION-400M       |   LiT - LAION-400M  |   SimZSS - COCO Captions - w/ bank |   SimZSS - COCO Captions - w/o bank  |  LiT - COCO Captions |
> |:---------------------------:|:-------------------:|:----------------------------------:|:------------------------------------:|:--------------------:|
> |38.9|37.5 |26.1 |25.5 |24.4 |

---

> > ### Author Response · Authors · 2024-11-20
> > **Reply part 2**
> >
> > ## Mining concepts from captions
> > The mining approach might indeed be suboptimal for noisier captions, such as those found in LAION-400M. However, the results of the experiment on COCO Captions without the concept bank suggest that SimZSS’s strong performance is primarily driven by the introduction of local supervision in the training objective, rather than by the breadth of concepts in the bank. Furthermore, the comparable performance of SimZSS and LiT on the zero-shot classification task supports the idea that the image-caption loss enables concept detection, while the concept-level loss enhances concept localization.
> >
> > In this study, we have used LAION-400M to demonstrate the versatility of the proposed method and its robustness to noisy datasets. Nonetheless, publicly available web-scale datasets such as [LAION-COCO](https://huggingface.co/datasets/laion/laion-coco), which include improved captions generated by pre-trained vision-language models, should be preferred. In such cases, the mining strategy remains applicable. Alternatively, using existing taxonomies like WordNet is also a viable option, as employing a large concept bank does not incur additional computational overhead.
> >
> > ## Conclusion
> > We hope we have addressed your concerns in a way that improves the paper's value in your view. If you have any further questions or feedback, please feel free to reach out, we would be glad to engage in a discussion.

---

> > > ### Comment · Reviewer_enbH · 2024-11-25
> > >
> > > I thank the authors for their response, which I find satisfactory. I also read the other reviews and found no plausible reason to change my initial positive feeling about the work. However, given the common misunderstanding of the concept bank, I would advise the authors to rename it and/or to better stress its purpose to guarantee a reader does not get confused. In addition, I believe the reviewers and the discussion may have outlined some limitations missing in the original work that should be included.

---

> > > > ### Author Response · Authors · 2024-11-26
> > > > **Response to additional feedback**
> > > >
> > > > Thank you for your positive response. We will consider this additional feedback regarding the naming of the concept bank in the camera-ready version. Furthermore, we plan to incorporate additional insights as suggested by reviewer wnrD.

---

### Official Review · Reviewer_wnrD · 2024-11-02

**Soundness:** 4
**Presentation:** 4
**Contribution:** 3
**Rating:** 6
**Confidence:** 4

**Summary:**

The paper introduces SimZSS, a simple framework that enables open-vocabulary zero-shot segmentation by leveraging pretrained vision-only models within a vision-language contrastive learning paradigm. Addressing the challenge of poor localization in dense tasks, SimZSS decouples visual representation learning from cross-modality alignment by utilizing frozen, spatially aware vision models like DINOv2 and focusing on training only the text encoder. The framework identifies local concepts within captions using linguistic cues and aligns them with corresponding visual concepts in images through similarity-based pooling.

**Strengths:**

- The paper is well-written and easy to follow. The authors provide clear explanations of their methodology. I would like to highlight the quality of the figures and tables in a visually pleasing way that enhances the understanding of the content.

- While the problem of localization in vision-language models is not novel, the proposed approach offers a novel perspective. By freezing the vision backbone and only training the text encoder, the authors leverage pretrained self-supervised models effectively, resulting in efficient training.

- The method demonstrates state-of-the-art performance on multiple zero-shot segmentation benchmarks.

**Weaknesses:**

1. It is surprising that the main paper lacks essential details about the training data and the pretrained models used. Given that the paper is only 9 pages (below the 10-page limit), including this information in the main text is necessary.

2.
(a) One major limitation is the use of a predefined concept bamk. The authors claim that it does not impact the breadth of the concept the model can localize(sec. 4.4). However, in Table 4, removing PascalVOC classes from the concept bank decreases the performance. Therefore, it seems that the breath of the concept bank is a crucial component. Also, the authors could evaluate on datasets with much diverse set of concepts e.g. OpenImagesV7 (which covers more than 5000 classes), this way the breadth of the vocabulary learned by the model can effectively be assessed.

(b) Moreover, to identify the recognition capabilities of the trained model, I would expect the authors to evaluate the performance of more classification datasets than just ImageNet. More specifically, it is common practice I think to evaluate on the set of 38 classification datasets (https://github.com/LAION-AI/CLIP_benchmark). Also, table 3 should include the performance of at least the OpenCLIP-ViT/16 trained on LAION-400M as the training dataset is the same, whose zero-shot accuracy on ImageNet is 67.05 vs 64.1 for your ViT-B/14 which indicates that your method hurts the overall recognition capability of the final model. This tradeoff between localization and recognition should be discussed in the paper.

(c)  Since the concept bank is derived from the union of class names from the segmentation datasets used for evaluation, the framework may be biased towards these classes. This design choice limits the model's ability to generalize to a truly open vocabulary, as it may not effectively localize concepts outside the predefined bank.

3. The paper misses important related work, particularly training-free approaches that tackle poor localization in VLMs from a different angle. For instance, methods like [a][b][c][d] achieve competitive performance without requiring additional training and can detect concepts not covered by a predefined concept bank. Including comparisons with such methods would provide a more comprehensive evaluation.

References:
[a] Lan, M., Chen, C., Ke, Y., Wang, X., Feng, L. and Zhang, W., 2024. Proxyclip: Proxy attention improves clip for open-vocabulary segmentation. arXiv preprint arXiv:2408.04883.

[b] Li, Y., Wang, H., Duan, Y. and Li, X., 2023. Clip surgery for better explainability with enhancement in open-vocabulary tasks. arXiv preprint arXiv:2304.05653.

[c] Bousselham, W., Petersen, F., Ferrari, V. and Kuehne, H., 2024. Grounding everything: Emerging localization properties in vision-language transformers. In Proceedings of the IEEE/CVF Conference on Computer Vision and Pattern Recognition (pp. 3828-3837).

[d] Zhou, C., Loy, C.C. and Dai, B., 2022, October. Extract free dense labels from clip. In European Conference on Computer Vision (pp. 696-712). Cham: Springer Nature Switzerland.

[e] Benenson, R. and Ferrari, V., 2022. From colouring-in to pointillism: revisiting semantic segmentation supervision. arXiv preprint arXiv:2210.14142.

**Questions:**

- Given that the predefined concept bank seems to impact performance, how does the model perform on datasets with a much broader set of concepts, such as OpenImagesV7, which covers over 5000 classes? Has the model been evaluated on classes outside of the concept bank, and if so, what are the results?

- Have you evaluated the recognition capabilities of the trained model on other classification datasets beyond ImageNet? Including evaluations on a wider range of datasets could provide deeper insights into how the method affects recognition performance.

-  How does your model compare with training-free approaches?

- Could you discuss the potential trade-off between improved localization and any degradation in recognition performance? Understanding this trade-off would help assess the overall impact of your method on different tasks.

---

> ### Author Response · Authors · 2024-11-19
> **Reply part 1**
>
> ## Overview
>
> We thank the reviewer for the detailed and constructive feedback as well as the appreciation of our work in terms of soundness, presentation and contribution. Below, we address your questions and concerns in detail.
>
> ## Lack of details about the training and data
> Based on your feedback, we have relocated the "Experimental Setup" subsection from the Appendix to the main text.
>
> ## Usage of a predefined concept bank & open-vocabulary capabilities
>
> There appears to be a general misunderstanding regarding the role of the concept bank. Its primary purpose is to filter out noun phrases that may not appear in the image, addressing noise in the captions of large-scale web datasets like LAION-400M. For more curated image-caption pair datasets, the concept bank is optional. To support this, we trained SimZSS without the concept bank. As expected, the scores are slightly lower compared to using the concept bank. However, the results consistently outperform the baseline (LiT), which focuses solely on enforcing consistency between captions and images representations. Moreover, we observe that the performance of SimZSS without the concept bank is closer to the version with the concept bank than to LiT. This demonstrates that SimZSS's performance is driven more by the consistency objective between noun phrases and localized image regions than by the concept bank itself. Considering the results achieved by concurrent baselines on the same tasks (Tables 1 and 2 in the main text), we conclude that SimZSS yields great performance compared to existing methods, irrespective of whether a concept bank is used, which shows that a concept does not need to appear in the concept bank to be properly localized at test time.
>
> #### Table 1: Zero-shot segmentation with and with out the bank of concepts on datasets without background.
> |  Method  |   Bank  |  Pascal VOC | Pascal Context | COCO-Stuff | Cityscapes | ADE20K |
> |---------:|--------:|------------:|---------------:|-----------:|-----------:|-------:|
> |  LiT     |     -   |        86.1 |  35.5          | 25.6       |  25.8      | 18.1   |
> |  SimZSS  |    w/o  |        89.2 |  41.1          | 27.6       |  31.2      | 20.0   |
> |  SimZSS  |    w/   |        90.3 |  43.1          | 29.0       |  33.0      | 21.8   |
>
>
> #### Table 2: Zero-shot segmentation with and with out the bank of concepts on datasets with background.
> |  Method  |   Bank  |  Pascal Context | COCO-Object | Pascal VOC |
> |---------:|--------:|------------:|---------------:|------------:|
> |  LiT     |     -   |        31.5 |  39.5          | 51.4        |
> |  SimZSS  |    w/o  |        35.2 |  42.6          | 56.8        |
> |  SimZSS  |    w/   |        37.2 |  43.5          | 58.4        |
>
> Regarding the open-vocabulary capabilities of SimZSS, we argue that any zero-shot segmentation method using free-form text as a prompt qualifies as open-vocabulary. Moreover, SimZSS differentiates itself from LiT by incorporating an additional loss at the concept level, complementing the cross-modal contrastive objective introduced by CLIP. Since both CLIP and LiT are regarded as open-vocabulary models, there is no reason to consider SimZSS differently, particularly in light of the results from the preceding experiment. Furthermore, in the following section, we demonstrate that the additional loss introduced does not compromise the zero-shot classification capability.

---

> > ### Author Response · Authors · 2024-11-19
> > **Reply part 2**
> >
> > ## Extensive classification benchmarks & tradeoffs between classification and localization
> >
> > Following the reviewer's suggestion we have evaluated the zero-shot classification of SimZSS using the recommended 38 datasets. In the tables below, we observe that in average SimZSS does **not** compromise the zero-shot classification accuracy compared to a LiT pre-training with the same backbone (ViT-B/14 pre-trained with DINOv2) and dataset. Similarly, we observe that using the bank of concepts does not decrease the average zero-shot classification performance, which contradicts the hypothesis that its usage restricts the breadth of concepts that can be detected/localized.
> >
> > It is important to note that we do not compare SimZSS with CLIP, as we lack access to comparable checkpoints. In contrast, for LiT, we have models trained under identical conditions, using the same backbones, datasets, and number of epochs. Notably, LiT serves as a stronger baseline than CLIP (see https://arxiv.org/pdf/2111.07991).
> >
> > The claim that SimZSS negatively impacts the overall recognition capability is not supported by evidence. While CLIP achieves a top-1 accuracy of 67.0% after **32 epochs** of training on LAION-400M, the text encoder in SimZSS is trained from scratch on LAION-400M for **only 1 epoch**, and a single epoch of CLIP is substantially more compute-intensive than one of SimZSS or LiT. Furthermore, the top-1 accuracy of CLIP after only 1 epoch on LAION-400M is approximately 43% (see second [figure](https://raw.githubusercontent.com/mlfoundations/open_clip/main/docs/laion_clip_zeroshot_b16.png) in the [Pretrained.md](https://github.com/mlfoundations/open_clip/blob/main/docs/PRETRAINED.md) README of [open_clip](https://github.com/mlfoundations/open_clip)).  Overall, the comparison with CLIP is not like-for-like, which is why we instead compare SimZSS with LiT.
> >
> >
> > #### Table 3: Average zero-shot classification top-1 accuracy.
> > |   SimZSS - LAION-400M       |   LiT - LAION-400M  |   SimZSS - COCO Captions - w/ bank |   SimZSS - COCO Captions - w/o bank  |  LiT - COCO Captions |
> > |:----------------------------:|:--------------------:|:-----------------------------------:|:-------------------------------------:|:---------------------:|
> > |                        38.9 |                37.5 |                               26.1 |                                 25.5 |                 24.4 |

---

> ### Author Response · Authors · 2024-11-19
> **Reply part 3**
>
> #### Table 4: Zero-shot classification top-1 accuracy.
> | dataset                                     |   SimZSS - LAION-400M |   LiT - LAION-400M  |   SimZSS - COCO Captions - w/ bank |   SimZSS - COCO Captions - w/o bank | LiT - COCO Captions |
> |:--------------------------------------------:|:----------------------:|:--------------------:|:-----------------------------------:|:------------------------------------:|:--------------------:|
> | wds/cars                                | 46.7 | 50   |  2.5 |  1.3 | 2   |
> | wds/country211                          |  7.1 |  7.4 |  0.7 |  0.7 | 0.6 |
> | wds/fer2013                             | 19.2 | 18.9 | 28.1 | 30.1 |25.4 |
> | wds/fgvc_aircraft                       |  7.4 |  8.5 |  2.2 |  1.7 | 2.2 |
> | wds/gtsrb                               |  7.2 |  7.6 |  4.6 |  4.1 | 4   |
> | wds/imagenet-a                          | 52.2 | 51.4 | 20   | 18.7 |18.6 |
> | wds/voc2007_multilabel                  | 37.7 | 38   | 15.3 | 14.2 |13.8 |
> | wds/imagenet-r                          | 71.6 | 70.7 | 35.7 | 35.3 |33.2 |
> | wds/imagenet1k                          | 64.1 | 63.6 | 24.3 | 24.4 |22.6 |
> | wds/imagenet_sketch                     | 50.3 | 49.4 | 19.7 | 19.3 |18.3 |
> | wds/imagenetv2                          | 57.5 | 57.4 | 22   | 21.6 |20   |
> | wds/mnist                               | 10.9 | 14   | 11.7 |  9.8 |13.4 |
> | wds/objectnet                           | 50.2 | 48.8 | 30.1 | 29.5 |28   |
> | wds/renderedsst2                        | 50   | 50   | 52.2 | 50.1 |49.9 |
> | wds/stl10                               | 93.9 | 94.7 | 92.4 | 96.5 |91.8 |
> | wds/sun397                              | 56.3 | 55.7 | 39.4 | 38.8 |38.7 |
> | wds/voc2007                             | 68.7 | 47.2 | 75.4 | 73.5 |73.8 |
> | wds/vtab/caltech101                     | 83.7 | 82.8 | 52.5 | 51.4 |50   |
> | wds/vtab/cifar10                        | 87   | 90.6 | 95.3 | 94.6 |94.5 |
> | wds/vtab/cifar100                       | 78   | 78.3 | 49.8 | 47.5 |46.3 |
> | wds/vtab/clevr_closest_object_distance  | 12.9 | 22.5 |  7.9 | 20.3 |22.2 |
> | wds/vtab/clevr_count_all                | 12   | 12.9 | 10.9 | 13.5 |12.5 |
> | wds/vtab/diabetic_retinopathy           | 63.9 | 14.6 | 53.9 | 49.4 |20.4 |
> | wds/vtab/dmlab                          | 17.9 | 17.7 | 14.9 | 14.3 |11.8 |
> | wds/vtab/dsprites_label_orientation     |  2.5 |  1.8 |  2.3 |  2   | 2.3 |
> | wds/vtab/dsprites_label_x_position      |  3   |  3.1 |  3.3 |  3.1 | 3.2 |
> | wds/vtab/dsprites_label_y_position      |  3.1 |  3.2 |  3   |  3.4 | 3   |
> | wds/vtab/dtd                            | 33.5 | 31.9 |  5.4 |  5.4 | 4.5 |
> | wds/vtab/eurosat                        | 29   | 19.1 | 11.4 | 22.1 |16.1 |
> | wds/vtab/flowers                        | 67.6 | 64.1 |  7.8 | 10.1 | 8.6 |
> | wds/vtab/kitti_closest_vehicle_distance |  9.4 | 34.3 | 26.4 | 16.6 |15.6 |
> | wds/vtab/pcam                           | 50.6 | 52.3 | 57.8 | 46.4 |59.8 |
> | wds/vtab/pets                           | 81.2 | 77.4 | 29.7 | 24.7 |27.1 |
> | wds/vtab/resisc45                       | 25.4 | 23.6 | 18.9 | 15.6 |16.4 |
> | wds/vtab/smallnorb_label_azimuth        |  6.4 |  5.1 |  5.2 |  6.3 | 5.8 |
> | wds/vtab/smallnorb_label_elevation      | 11   | 10.2 | 11.4 | 11.6 |12.6 |
> | wds/vtab/svhn                           | 10.5 |  9.4 | 20.1 | 14.2 |13.4 |
>
>
> ## Missing related works
>
> Thank you for pointing the missing entries from the related works. We recognize the relevance of these works in this context and will include these in the updated draft that will be uploaded before the end of
> the discussion period. Furthermore, we will add a subsection specifically dedicated to the comparison with training-free approaches.
>
> ## Conclusion
> We hope we've effectively addressed your concerns and encouraged a reconsideration of your rating. If you have any additional questions or feedback, please don't hesitate to reach out, we're happy to engage in further discussion.

---

> > ### Comment · Reviewer_wnrD · 2024-11-25
> > **Response to Author Rebuttal**
> >
> > I appreciate the detailed response and the additional experiments provided. The clarifications have addressed several of my concerns:
> > - Moving the experimental setup details to the main paper and adding comparisons to training-free methods will significantly improve the manuscript's completeness.
> > - The additional experiments without the concept bank are valuable and demonstrate that SimZSS can maintain good performance even without this component, though with some performance degradation.
> >
> > However, I still have some reservations about certain aspects of the response:
> > ## Classification Performance and Comparisons
> > The argument against comparing with OpenCLIP is not entirely convincing. While training conditions may differ, comparison with state-of-the-art models is essential for proper evaluation. Some key points:
> > - The fact that SimZSS achieves 38.9% average top-1 accuracy compared to OpenCLIP-ViT-B-16's 56.21% (on LAION-400M) suggests a significant trade-off between localization and recognition capabilities.
> > - The comparison with a 1-epoch CLIP (43% accuracy) isn't particularly relevant since CLIP trains both encoders from scratch, while SimZSS benefits from a pre-trained DINOv2 backbone.
> > - If computational efficiency is a feature of SimZSS, this should be explicitly discussed as a trade-off rather than used to avoid comparisons with longer-trained models.
> >
> > ## Open-Vocabulary Capabilities
> > Regarding the open-vocabulary nature of the model, it's important to note that this is not a binary property but rather exists on a spectrum. While SimZSS is indeed open-vocabulary by design, the extent of its vocabulary breadth remains somewhat unclear. For instance:
> > - The ability to recognize broad categories (e.g., "car") doesn't necessarily translate to fine-grained distinctions (e.g., between an SUV and a van)
> > - Additional evaluation on datasets with more diverse concept sets would help quantify the breadth of the model's vocabulary
> > ## Conclusion
> > Despite these concerns, I acknowledge the valuable contributions of this work:
> > - The novel approach to zero-shot segmentation
> > - The efficient training pipeline
> > - The strong performance on various benchmarks
> >
> > Given these strengths and the authors' thorough response, I am increasing my score to 6. The work presents interesting ideas and achieves reasonable performance, though I encourage the authors to address the remaining concerns about comparative evaluation and vocabulary breadth in the final version.

---

> > > ### Author Response · Authors · 2024-11-26
> > > **Response to additional feedback**
> > >
> > > Thank you for acknowledging the valuable contributions of our work as well as for providing additional feedback once again.
> > >
> > > ## Classification Performance and Comparisons
> > >
> > > Figure 1 in the LiT [paper](https://arxiv.org/pdf/2111.07991) demonstrates that LiT consistently outperforms CLIP independent of the number of training epochs. Our approach builds upon LiT by introducing an additional localized objective. To ensure that this added term does not compromise the recognition performance, we compared the classification performance of SimZSS against that of LiT trained in the exact same setting. The results confirm that the localized objective does not trade recognition performance for localization performance.
> > >
> > > We acknowledge that an epoch-based comparison with CLIP is not entirely fair, given our use of a pretrained vision encoder, which is why we initially omitted such a comparison. However, to further support our claims, we plan to train both SimZSS and LiT for multiple epochs on LAION-400M (with $1 < \text{epoch} \leq 32$) and include these results alongside those from CLIP in Table 3 of the camera-ready version.
> > >
> > >
> > > ## Open-Vocabulary Capabilities
> > >
> > > We agree that evaluating on datasets with more diverse concept sets would provide a better assessment of the learned vocabulary's breadth. Such an evaluation should be conducted not only for SimZSS but also for all other methods in the literature.
> > >
> > > ## Conclusion
> > >
> > > We aim to address the above points in the camera-ready version and hope these results will be beneficial for the community.

---

### Official Review · Reviewer_88HB · 2024-11-03

**Soundness:** 3
**Presentation:** 3
**Contribution:** 3
**Rating:** 6
**Confidence:** 4

**Summary:**

The paper introduces SimZSS, a straightforward yet effective framework for zero-shot semantic segmentation. This framework aims to align vision encoders, which exhibit spatial awareness, with textual descriptions. SimZSS leverages text concept representations to extract corresponding visual representations based on similarity metrics. It ensures cross-modality consistency by employing different objectives at the sample and concept levels. Notably, the method requires minimal hyperparameters and does not rely on the supervision of semantic masks. Additionally, SimZSS can be easily adapted to various backbones and datasets. Overall, the method achieves state-of-the-art results across standard zero-shot segmentation benchmarks.

**Strengths:**

1. The paper is easy to understand and follow.
2. The method is straightforward, easy to implement, and can be readily adapted to various backbones and training datasets.
3. The method can be trained without the supervision of semantic masks, reducing the burden of annotations.
4. The motivation for proposing a concept-level objective is clear and more suitable compared to a contrastive objective in scenarios where concepts encode individual objects that are likely to occur multiple times within a batch.

**Weaknesses:**

1. It is unclear how the final segmentation masks are generated during inference. Is there a similarity threshold used to determine the class names to which visual tokens belong? If so, how does the performance vary with different threshold settings?
2. There is a lack of analysis explaining why SimZSS outperforms other zero-shot semantic segmentation methods.

**Questions:**

1. Since the visual encoder is frozen, the quality of the segmentation masks heavily depends on the encoder's ability to recognize spatial positions. Would the performance improve if the visual encoder were fully or partially fine-tuned?
2. In an extreme scenario, how significantly would the test performance be affected if the class names in the training set and test set are entirely different?

---

> ### Author Response · Authors · 2024-11-20
> **Reply part 1**
>
> ## Overview
> We thank the reviewer for the detailed and constructive feedback. Below, we address your questions and concerns in detail.
>
> ## Explanation behind SimZSS good performance
> ### Visual representation learning vs vision-language alignment
> As mentioned in Section 1 and identified by LiT, the cross-modal contrastive
> objective used in CLIP is a suboptimal approach for learning visual
> representations, both in terms of efficiency and effectiveness. Achieving zero-shot segmentation relies on two key requirements:
>
> - The vision encoder must be capable of clustering groups of pixels
>   with shared semantics.
> - The text encoder must be sufficiently aligned with the visual modality to
>   accurately label these clusters.
>
> We argue that the second task is significantly easier to accomplish than the
> first and that the first task can be addressed independently of the second. This
> is the main reason behind the strong performance we report. Indeed, most
> concurrent methods begin with pre-aligned vision and text encoders (task 2) and
> attempt to transfer image-level cross-modal alignment to pixel-level alignment.
> As shown in the following tables, CLIP demonstrates limited cross-modal alignment at finer levels of granularity.
>
> #### Table 1: Zero-shot segmentation with a ViT-B/16 pre-trained with CLIP on LAION-400M for 32 epochs on datasets without background.
> |  Method   |  Pascal VOC  | Pascal Context  | COCO-Stuff  | Cityscapes  | ADE20K  |
> |:---------:|:------------:|:---------------:|:-----------:|:-----------:|:-------:|
> |   CLIP    |     35.1     |       7.7       |     4.2     |     1.8     |   2.0   |
>
> #### Table 2: Zero-shot segmentation with a ViT-B/16 pre-trained with CLIP on LAION-400M for 32 epochs on datasets with background.
> |  Method   | Pascal Context |   COCO-Object   |  Pascal VOC  |
> |:---------:|:--------------:|:---------------:|:------------:|
> |   CLIP    |      6.9       |       5.2       |     11.1     |
>
> ### Finetuning/freezing the visual encoder
> As opposed to previous methods relying on CLIP (which does not exhibit good dense vision-language alignment), our approach starts with a pre-trained vision encoder that
> demonstrates semantic coherence at the pixel level, and we focus on aligning a text encoder to it. This is illustrated in Table 11 of the main text. We show that the same vision backbone (ViT-B/14 pre-trained with DINOv2) equipped with a linear layer can achieve over 40 mIoU on ADE20K. Zero-shot segmentation can be interpreted as classification with a frozen linear layer that is initialized with embeddings of textual labels. This result highlights that with a better alignment of the text encoder, SimZSS could reach over 40 mIoU on ADE20K, showcasing that the quality of the vision encoder is far from being the bottleneck.
>
> This design choice is further supported by the strong performance of LiT on zero-shot segmentation tasks. The remaining performance gap can be attributed to SimZSS's ability to identify concept pairs across modalities
> and enforce consistency between them. Specifically, the consistency objective is
> implemented via a linear projection layer, where the weights are derived by
> summing the concept representations (of the same concept) within the batch, combined with a
> cross-entropy loss. As such, the training of the concept-level loss is completely aligned with the way classification is done at inference time, further explaining the observed strong performance.
>
> We will make this analysis explicitly clear in the paper, add an entry for CLIP zero-shot segmentation in Table 1 and 2 of the main text, and add pointers to Table 11 which is located in the Appendix.

---

> > ### Author Response · Authors · 2024-11-20
> > **Reply part 2**
> >
> > ## Impact of the concept bank
> >
> > There appears to be a general misunderstanding regarding the role of the concept bank. Its primary purpose is to filter out noun phrases that may not appear in the image, addressing noise in the captions of large-scale web datasets like LAION-400M. For more curated image-caption pair datasets, the concept bank is optional. To support this, we trained SimZSS without the concept bank. As expected, the scores are slightly lower compared to using the concept bank. However, the results consistently outperform the baseline (LiT), which focuses solely on enforcing consistency between captions and images representations. Moreover, we observe that the performance of SimZSS without the concept bank is closer to the version with the concept bank than to LiT. This demonstrates that SimZSS's performance is driven more by the consistency objective between noun phrases and localized image regions than by the concept bank itself. Considering the results achieved by concurrent baselines on the same tasks (Tables 1 and 2 in the main text), we conclude that SimZSS yields great performance compared to existing methods, irrespective of whether a concept bank is used, which shows that a concept does not need to appear in the concept bank to be properly localized at test time.
> >
> > #### Table 3: Zero-shot segmentation with and with out the bank of concepts on datasets without background.
> > |  Method   |   Bank   |  Pascal VOC  | Pascal Context  | COCO-Stuff  | Cityscapes  | ADE20K  |
> > |:---------:|:--------:|:------------:|:---------------:|:-----------:|:-----------:|:-------:|
> > |    LiT    |    -     |     86.1     |      35.5       |    25.6     |    25.8     |  18.1   |
> > |  SimZSS   |   w/o    |     89.2     |      41.1       |    27.6     |    31.2     |  20.0   |
> > |  SimZSS   |    w/    |     90.3     |      43.1       |    29.0     |    33.0     |  21.8   |
> >
> >
> > #### Table 4: Zero-shot segmentation with and with out the bank of concepts on datasets with background.
> > |  Method   |   Bank   | Pascal Context |   COCO-Object   |  Pascal VOC  |
> > |:---------:|:--------:|:--------------:|:---------------:|:------------:|
> > |    LiT    |    -     |      31.5      |      39.5       |     51.4     |
> > |  SimZSS   |   w/o    |      35.2      |      42.6       |     56.8     |
> > |  SimZSS   |    w/    |      37.2      |      43.5       |     58.4     |
> >
> >
> > ## Generation of the final masks during inference
> > At inference time, images are processed through the vision tower to obtain dense visual representations, which are then projected onto the class descriptions embedded by the text encoder. When the class descriptions comprehensively cover all pixels in the image (as is the case with COCO-Stuff, for instance), each pixel is assigned to the class it most closely resembles using an argmax operation.
> >
> > In contrast, when the set of classes is incomplete and some pixels/patches are labeled as ``background``, we rely on the prediction confidence for the other classes, combined with a threshold, to determine whether a pixel belongs to the background or one of the foreground classes. By design, setting the threshold to 1.0 (after softmax normalization) assigns all pixels to the background, whereas setting it to 0.0 results in no pixels predicted as ``background``. Naturally, this threshold impacts the overall score, regardless of the pre-training method, model, or evaluation dataset.
> >
> > We acknowledge that this approach is simplistic and adopt it solely to remain consistent with prior work, even though some baselines use more sophisticated techniques for background detection. Generally, we would advocate for using a more comprehensive set of classes and then designating certain classes as background a posteriori, or focusing the evaluation on datasets where every pixels are annotated, such as COCO-Stuff.
> >
> > ## Conclusion
> > We hope we have addressed your concerns effectively, improving the paper's value from your perspective. If you have any additional we would be happy to discuss this further.

---

> > > ### Comment · Reviewer_88HB · 2024-11-25
> > > **Official Comment by Reviewer 88HB**
> > >
> > > Thank you for the detailed response. It clarifies most of my concerns, and I will keep my original rating, leaning toward to acceptance.

---

> > > > ### Author Response · Authors · 2024-11-26
> > > >
> > > > Thank you for your positive reply.

---

### Official Review · Reviewer_t3xp · 2024-11-05

**Soundness:** 2
**Presentation:** 2
**Contribution:** 2
**Rating:** 5
**Confidence:** 4

**Summary:**

This work proposes a simple framework for open-vocabulary Zero-Shot Segmentation with two key principles of leveraging frozen
vision-only models and exploiting the discrete nature of text and linguistic knowledge.

**Strengths:**

The experimental performance is good.

**Weaknesses:**

1. The novelty and contribution is limited. First, I prefer simple method. I believe that the simple method is more valuable for applications and research. However, simple method requires more deep analysis and insights. Unfortunately, this work only presents a simple method without any insights. The method is designed without motivation or explanation. This paper looks like a experiment report, rather than a research paper. For a good research paper, the authors need to tell new insights, rather than just propose a model and conduct some experiments.
2. The comparison is unfair. The proposed method is trained on COCO Captions (Lin et al., 2014; Chen et al., 2015) and LAION-400M (Schuhmann et al., 2021) and the image encoder is pretrained on LVD-142M dataset, as shown in L764-L774. However, other methods are not trained on these large-scale dataset. Even trained on these large datasets, the performance improvement is limited and the proposed method even performs worse than previous methods on Pascal VOC dataset. For example, the proposed method SimZSS trained on LAION-400M achieves 48.6 on Pascal VOC, which is much lower than CLIP-DINOiser (Wysoczanska et al., 2023), CLIP-DIY (Wysoczanska et al., 2024), OVSegmentor (Xu et al., 2023a), OVDiff (Karazija et al., 2023), CLIPpy (Ranasinghe et al., 2023) and TCL (Cha et al., 2023).

Overall, this work propose a simple method without motivation and experimental analysis for the working mechanism. The good performance is mainly derived from its large training dataset.

**Questions:**

What is the motivation of the proposed method?
What is the reason of the good performance? I think it is because the large training dataset.

---

> ### Author Response · Authors · 2024-11-19
> **Reply part 1**
>
> We thank you for taking the time to review our paper. We address all your points below:
>
> > The comparison is unfair. The proposed method is trained on COCO Captions (Lin
> et al., 2014; Chen et al., 2015) and LAION-400M (Schuhmann et al., 2021) and the
> image encoder is pretrained on LVD-142M dataset, as shown in L764-L774. However,
> other methods are not trained on these large-scale dataset. Even trained on
> these large datasets, the performance improvement is limited.
>
> This statement is incorrect. In fact, methods such as CLIP-DINOiser, CLIP-DIY,
> and TCL utilize models pre-trained with CLIP on web-scale datasets like LAION-2B
> for **multiple epochs** (see https://github.com/mlfoundations/open_clip).
> Similarly, CLIPpy is pre-trained on HQITP-134M, a dataset containing 134 million
> samples. For OVDiff, we report results obtained using DINO+CLIP+StableDiffusion
> pre-trained image encoders, where both CLIP and StableDiffusion models are
> trained on web-scale datasets. Moreover, OVSegmentor employs a DINO pre-trained image encoder and a BERT pre-trained text encoder which was pretrained with BooksCorpus (800 million words) (Zhu et al., 2015) and English Wikipedia (2,500 million words). Additionally, they perform alignment on CC4M, a dataset with 4 million samples.
>
> We would like to emphasize that assigning a confidence score of 5 necessitates a deep familiarity with related works, which does not appear to be reflected in this assessment.
> We have the impression the reviewer is not well aware of the literature on this topic and therefore may not be well placed to review our paper.
>
> > The novelty and contribution is limited.
>
> We would be grateful if you could point us toward any existing zero-shot
> segmentation works that significantly overlap with SimZSS.
>
> > First, I prefer simple method. I believe that the simple method is more
> valuable for applications and research. However, simple method requires more
> deep analysis and insights. Unfortunately, this work only presents a simple
> method without any insights. The method is designed without motivation or
> explanation. This paper looks like a experiment report, rather than a research
> paper. For a good research paper, the authors need to tell new insights, rather
> than just propose a model and conduct some experiments.
>
> We believe that scientifically driven insights are best derived through rigorous
> experimentation, which is why we meticulously designed a comprehensive set of
> experiments, each accompanied by detailed conclusions. We would appreciate it if you could highlight specific aspects of SimZSS where additional insights or
> explanations might be needed.
>
> > The proposed method even performs worse than previous methods on Pascal VOC
> dataset. For example, the proposed method SimZSS trained on LAION-400M achieves
> 48.6 on Pascal VOC, which is much lower than CLIP-DINOiser (Wysoczanska et al.,
> 2023), CLIP-DIY (Wysoczanska et al., 2024), OVSegmentor (Xu et al., 2023a),
> OVDiff (Karazija et al., 2023), CLIPpy (Ranasinghe et al., 2023) and TCL (Cha et
> al., 2023).
>
> The datapoint you selected is only one of the many evaluation dataset / alignment dataset combination we report in the paper. If you had picked our best performing setting (i.e. alignment on COCO-Captions), you would observe that only OVDiff (Karazija et al., 2023), CLIP-DIY (Wysoczanska et al., 2024) and CLIP-DINOiser (Wysoczanska et al., 2023) outperform it. We also provide insights in the paper regarding this (see L395 before paper update).
>
> Overall, we outperform all other methods on 7 out of 8 evaluations and all other methods on average over these 8 evaluations. The aggregate performance of SimZSS is superior to all other baselines which is what we state in the paper.

---

> > ### Author Response · Authors · 2024-11-19
> > **Reply part 2**
> >
> > > What is the motivation of the proposed method? What is the reason of the good
> > performance? I think it is because the large training dataset.
> >
> > As mentioned in Section 1 and identified by LiT, the cross-modal contrastive
> > objective used in CLIP is a suboptimal approach for learning visual
> > representations, both in terms of efficiency and effectiveness. Achieving
> > zero-shot classification or segmentation relies on two key requirements:
> >
> > - The vision encoder must be capable of clustering images or groups of pixels
> >   with shared semantics.
> > - The text encoder must be sufficiently aligned with the visual modality to
> >   accurately label these clusters.
> >
> > We argue that the second task is significantly easier to accomplish than the
> > first and that the first task can be addressed independently of the second. This
> > is the main reason behind the strong performance we report. Indeed, most
> > concurrent methods begin with pre-aligned vision and text encoders (task 2) and
> > attempt to transfer image-level cross-modal alignment to pixel-level alignment.
> > As shown in the following tables, CLIP demonstrates limited cross-modal alignment at finer levels of granularity.
> >
> > #### Table 1: Zero-shot segmentation with a ViT-B/16 pre-trained with CLIP on LAION-400M for 32 epochs on datasets without background.
> > |  Method  |  Pascal VOC | Pascal Context | COCO-Stuff | Cityscapes | ADE20K |
> > |---------:|------------:|---------------:|-----------:|-----------:|-------:|
> > |  CLIP    |        35.1 |  7.7           | 4.2        |   1.8      |  2.0   |
> >
> > #### Table 2: Zero-shot segmentation with a ViT-B/16 pre-trained with CLIP on LAION-400M for 32 epochs on datasets with background.
> > |  Method  |  Pascal Context | COCO-Object | Pascal VOC |
> > |---------:|------------:|---------------:|------------:|
> > |  CLIP    |        6.9  |  5.2           | 11.1        |
> >
> > In contrast, our approach starts with a pre-trained vision encoder that
> > demonstrates semantic coherence at the pixel level, and we focus on aligning a
> > text encoder to it.
> >
> > As discussed above, there is no evidence supporting the claim that our
> > performance hinges on larger training datasets. In fact, our results indicate
> > the opposite: we achieve better outcomes using COCO Captions, a dataset several
> > orders of magnitude smaller than LAION-400M.
> >
> > ### Conclusion
> >
> > We recognize that the motivation might not have been entirely clear to you and hope this response has clarified it. As we explained above, the review contains factual errors. We hope that, now that these are clarified and the motivation behind our design choices are better explained, the reviewer can better appreciate our contributions and we can engage in a constructive discussion. We are open to any actionable feedback. Thank you!

---

> ### Comment · Reviewer_t3xp · 2024-12-01
> **Official Comment by Reviewer t3xp**
>
> Thanks for your reply!
> The revised version is better.
> However, I disagree that the motivation is only derived from experimental improvement.
> The authors should provide convincing evidence to support their argument, rather than just saying the performance is improvement. The behind reason of improved performance can be diverse.
> Besides, the following statement is overclaimed:
> ```
> The vision encoder must be capable of clustering images or groups of pixels with shared semantics.
> The text encoder must be sufficiently aligned with the visual modality to accurately label these clusters.
> ```
>
> So I change my rating to 5.

---

> > ### Author Response · Authors · 2024-12-02
> >
> > Thank you for your response and for positively revising your score.
> >
> > The statement outlines two necessary conditions for achieving zero-shot open-vocabulary segmentation without additional supervision:
> >
> > - **Grouping of local representations:** For zero-shot segmentation to succeed, the local representations of similar objects must be clustered together in the embedding space. If this grouping does not occur, achieving good segmentation performance is impossible, regardless of the quality of the text encoder. This is because, during inference, the text encoder’s sole role is to generate the weights for the linear classifier.
> >
> > - **Alignment of text and vision modalities:** The text encoder must be well-aligned with the vision modality. Without such alignment, the embeddings of a class's textual description will not closely correspond to the local representations of pixels or patches associated with that class. Consequently, the resulting linear classifier will lack discriminative power, leading to poor zero-shot segmentation performance.
> >
> > Importantly, the revised version includes additional evidence supporting our motivation for separating these tasks. Specifically, Tables 1 and 2 now report the performance of CLIP on the zero-shot segmentation task, highlighting that jointly training the text and vision encoders is suboptimal for achieving fine-grained cross-modality alignment.
> >
> > Furthermore, Section A.3 of the revised draft provides evidence showing that the quality of the vision encoder is not the primary performance bottleneck of SimZSS, further justifying the decoupling of these tasks and the freezing of the vision backbone. Specifically, Table 8 shows the segmentation performance with a supervised linear layer (instead of the one initialized with the embedded text concepts) which strongly outperforms any open-vocabulary zero-shot method. This shows that using a frozen vision encoder is indeed a well justified choice. In the same section, we also discuss the limitations of the visual representations learned from the CLIP objective.
> >
> > We hope this response addresses any remaining concerns and provides the clarity needed to support a favorable evaluation of our work.

---

### Author Response · Authors · 2024-11-25
**Summary of discussion period**

Dear Reviewers,

Thank you for your initial feedback, individual responses, and the overall positive reassessment of the ratings.

Additionally, we have updated the manuscript to incorporate your suggestions. Key changes include (highlighted in blue):

- **Revised introduction:** The introduction has been rewritten to more effectively motivate the design choices behind SimZSS and explain how these decisions contribute to its strong performance.
- **Experimental setup section:** The main paper now includes a dedicated "Experimental Setup" sub-section, detailing the datasets and pre-trained models utilized.
- **CLIP performance:** We have added the performance of CLIP on the zero-shot segmentation benchmark and included additional insights to contextualize SimZSS's strong results.
- **Expanded classification benchmark:** The zero-shot classification benchmark now encompasses 38 commonly used datasets to provide a more comprehensive evaluation.
- **New experiment on concept bank:** We present a novel experiment examining the role of the concept bank, showing that SimZSS can generalize to concepts outside the bank and achieve excellent performance even without it.
- **Comparison with training-free approaches:** The Appendix now includes a new sub-section dedicated to comparing SimZSS with training-free methods.

We hope that these updates further enhance the value of the paper.

---

### Meta-Review · Area_Chair_1Ami · 2024-12-21

**Metareview:**

The paper presents a framework that aligns vision and text representations using a pretrained visual encoder, by incorporating the discrete nature of text modality and cross-modal consistency. It received three positive evaluations (two 6s and an 8) and one negative rating (5). The authors' rebuttal effectively maintained the positive scores and slightly improved the negative score to 5. Although there is still a minor concern from the negative reviewer regarding the motivation behind the study, this issue appears relatively minor. The authors have successfully addressed most other concerns. Consequently, the AC recommends acceptance of the paper.

**Additional Comments On Reviewer Discussion:**

The primary concerns raised were related to 1) the motivation behind the study, 2) the lack of analyses and explanations, 3) incomplete details regarding experimental setups, and 4) the necessity for a predefined concept bank. The authors responded by providing additional explanations and results, which the reviewers recognized as effectively resolving most of their concerns. However, the negative reviewer still contends that the motivation requires further justification, arguing that it primarily stems from empirical improvements—a claim the AC considers subjective.

---

### Decision · Program_Chairs · 2025-01-22

Accept (Poster)